# The human amniotic epithelium confers a bias to differentiate toward the neuroectoderm lineage in human embryonic stem cells

Daniela Ávila-González[1,2]*, Wendy Portillo[3], Carla P Barragán-Álvarez[2], Georgina Hernandez-Montes[4], Eliezer Flores-Garza[5], Anayansi Molina-Hernández[1], Néstor Emmanuel Díaz-Martínez[2], Néstor F Díaz[1]*

[1]Instituto Nacional de Perinatología, Mexico City, Mexico; [2]Laboratorio de Reprogramación Celular y Bioingeniería de Tejidos, Biotecnología Médica y Farmacéutica, Centro de Investigación y Asistencia en Tecnología y Diseño del Estado de Jalisco, Guadalajara, Mexico; [3]Instituto de Neurobiología, Universidad Nacional Autónoma de México, Querétaro, Mexico; [4]Red de Apoyo a la Investigación, Coordinación de la Investigación Científica, Universidad Nacional Autónoma de México e Instituto Nacional de Ciencias Médicas y Nutrición Salvador Zubirán, Mexico City, Mexico; [5]Departamento de Biología Molecular y Biotecnología, Instituto de Investigaciones Biomédicas, Universidad Nacional Autónoma de México, Mexico City, Mexico

*For correspondence:
avila.dela@gmail.com (DÁ-G);
nfdiaz00@yahoo.com.mx (NFD)

**Competing interest:** The authors declare that no competing interests exist.

**Abstract** Human embryonic stem cells (hESCs) derive from the epiblast and have pluripotent potential. To maintain the conventional conditions of the pluripotent potential in an undifferentiated state, inactivated mouse embryonic fibroblast (iMEF) is used as a feeder layer. However, it has been suggested that hESC under this conventional condition (hESC-iMEF) is an artifact that does not correspond to the in vitro counterpart of the human epiblast. Our previous studies demonstrated the use of an alternative feeder layer of human amniotic epithelial cells (hAECs) to derive and maintain hESC. We wondered if the hESC-hAEC culture could represent a different pluripotent stage than that of naïve or primed conventional conditions, simulating the stage in which the amniotic epithelium derives from the epiblast during peri-implantation. Like the conventional primed hESC-iMEF, hESC-hAEC has the same levels of expression as the 'pluripotency core' and does not express markers of naïve pluripotency. However, it presents a downregulation of HOX genes and genes associated with the endoderm and mesoderm, and it exhibits an increase in the expression of ectoderm lineage genes, specifically in the anterior neuroectoderm. Transcriptome analysis showed in hESC-hAEC an upregulated signature of genes coding for transcription factors involved in neural induction and forebrain development, and the ability to differentiate into a neural lineage was superior in comparison with conventional hESC-iMEF. We propose that the interaction of hESC with hAEC confers hESC a biased potential that resembles the anteriorized epiblast, which is predisposed to form the neural ectoderm.

## Editor's evaluation

A number of different stages of pluripotency have been described in the literature. Ávila-González et al., show that human amniotic epithelial cells (hAECs) can be used to support a unique primed pluripotent state in which cells are biased towards an anterior neural identity. The authors explore

the unique identity of these lineage-primed pluripotent cells suggesting they could be used for more efficient neural differentiation. The ability of hAECs to support a pre-patterned cell type that continues to express pluripotency markers also implies new roles for the aminion in patterning and supporting early human embryos.

## Introduction

Pluripotency is an ephemeral stage in early development in which the embryonic cells can differentiate into all the lineages that will constitute an organism. In recent years, studies have described naïve and primed pluripotency representing the pre- and post-implantation epiblast, respectively. Both states can be captured in vitro, adapting the embryonic cells to grow under defined conditions. Thus, mouse embryonic stem cells (mESC) and epiblast stem cells (EpiSC) arise as the in vitro counterparts of the naïve and primed identity in the mouse embryo, respectively (*Boroviak et al., 2014*; *Brons et al., 2007*). As regards humans, conventional conditions to maintain human pluripotent stem cells (hPSCs) in vitro involve the knockout serum replacement (KOSR) and Fibroblast Growth Factor 2 (FGF2) on the inactivated mouse embryonic fibroblast (iMEF) feeder layer (*Ávila-González et al., 2016*; *Ávila-González et al., 2021*). Hence, conventional hPSC are considered as the in vitro analogous of human primed pluripotency. However, their transcriptional state is equivalent to EpiSC and mouse post-implantation epiblast instead of human blastocyst. These data suggest that conventional hPSC could be an artifact (*Yang et al., 2019*).

On the other hand, the mouse epiblast exhibits a regional pluripotency spectrum. The first cells to exit from the naïve state are localized in the anterior epiblast to generate the epiblast-ectoderm at E6.5. In contrast, the posterior epiblast's pluripotent cells initiate the epithelial-mesenchymal transition to produce the streak primitive and primordial germ cell specification. Thus, pluripotency is a temporal process during development and has a spatial component that defines the anterior and posterior epiblast (*Peng et al., 2019*). Indeed, hPSC can self-organize to form anteriorized or posteriorized embryonic-like sacs capable of recapitulating pattern events inherent to the anterior-posterior epiblast (*Zheng, 2019*). These findings support the notion that there are different regionalized human pluripotent states analogous to mouse epiblast that could be captured in vitro, representing the plethora of pluripotent stem cells during *Homo sapiens* development. Lastly, amniotic epithelium segregation occurs directly from the epiblast during implantation in primates (*Boroviak and Nichols, 2017*), lacking process in murine development. Indeed, experiments where human embryos replicate implantation in vitro have confirmed that the human amniotic epithelium derives from the epiblast (*Deglincerti et al., 2016*; *Shahbazi et al., 2016*). Hence, pluripotent cells interaction with the amniotic epithelium could be crucial to understanding the transition among different pluripotency states and the regionalization in the human epiblast. Here, we explore whether the human embryonic stem cell (hESC)-human amniotic epithelial cell (hAEC) co-culture could exhibit a pluripotent substate with a different potential from the primed or naïve. We reported that hESC-hAEC shares the basal pluripotent signature analogous to the hESC-iMEF conventional primed state without naïve pluripotent-related factor expression. However, hAEC interaction provides a unique molecular profile and differentiation potential similar to that of the anterior epiblast hESC in vitro.

## Results

### hAEC and their conditioned media support undifferentiated hESC

We previously reported the hESC line (Amicqui-1) derivation on hAEC using poor-quality embryos as a source, demonstrating its pluripotency by teratoma assay and lineage-specific marker detection (*Ávila-González et al., 2015*). Our first question was if this condition confers a pluripotent state similar or different to the conventional condition (AMIQ hESC-iMEF) (*Figure 1A*). First, we compared POU5F1, SOX2, and NANOG expression in both conditions. Since hAEC have a 'pluripotency core' because of their epiblast origin (*Deglincerti et al., 2016*; *Shahbazi et al., 2016*; *Avila-González et al., 2019*; *García-Castro et al., 2015*), we included an hAEC sample to discard that the expression levels in the hESC-hAEC co-culture were partially due to the amniotic epithelium. Short tandem repeat (STR) genotyping was carried out to verify the authenticity of human cell lines (Amicqui-1, H1,

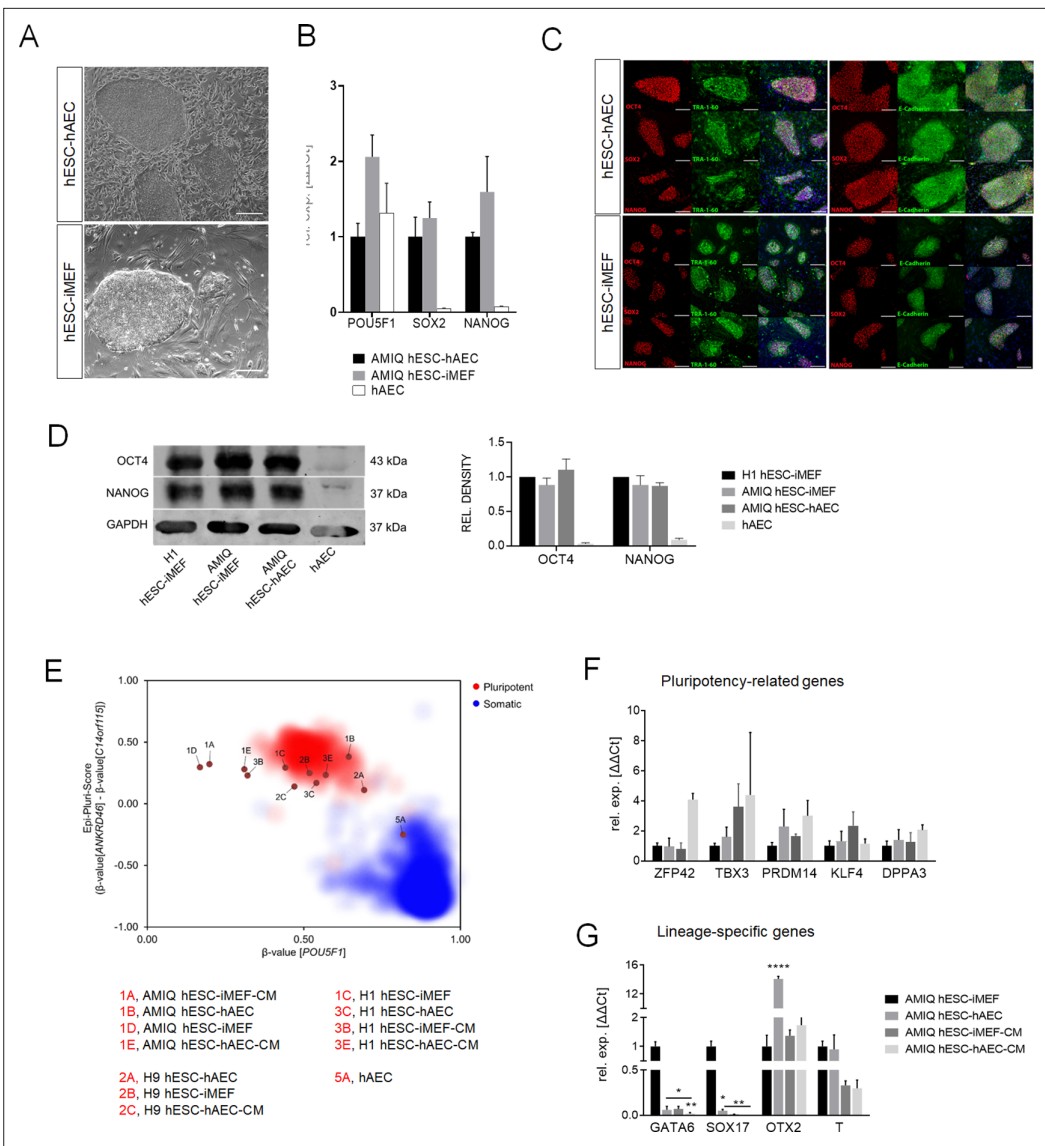

**Figure 1.** Characterization of pluripotency in alternative human embryonic stem cell (hESC)-human amniotic epithelial cell (hAEC) and conventional hESC-inactivated mouse embryonic fibroblast (iMEF). (**A**) Representative phase-contrast micrographs of Amicqui-1 hESC on hAEC feeder layer (AMIQ hESC-hAEC) and Amicqui-1 hESC on iMEF feeder layer (AMIQ hESC-iMEF). Scale bar, 200 µm. (**B**) Expression of the 'core pluripotency' genes in AMIQ hESC-hAEC, AMIQ hESC-iMEF, and hAEC alone by quantitative Polymerase Chain Reaction (qPCR). Data are mean ± SE, n=three biological samples per group, and three repetitions per sample. (**C**) Representative epifluorescence micrographs of AMIQ hESC-iMEF and AMIQ hESC-hAEC colonies with double immunostaining for each OCT4, SOX2, and NANOG transcription factor with TRA-1–60 surface antigen or E-cadherin cell-cell adhesion molecule. Scale bar, 50 µm. (**D**) Detection and semiquantitative analysis of OCT4 and NANOG proteins in H1 hESC-iMEF, AMIQ hESC-iMEF, AMIQ hESC-hAEC, and HAEC alone by western blot. n=three biological samples per group. (**E**) Epi-Pluri-Score reveals that hESC lines are pluripotent regardless of different culture conditions. DNA methylation (DNAm) was analyzed at three specific CpG sites. One of these CpGs was localized within the pluripotency-associated gene POU5F1 (also known as OCT4). Furthermore, the difference in DNAm levels (β-values) of CpGs in ANKRD46 and C14orf115 was determined and combined as Epi-Pluri-Score (***Lenz et al., 2015***). The red and blue clouds refer to DNAm profiles (all Illumina HumanMethylation27 BeadChip platform) of 264 pluripotent and 1951 non-pluripotent cell preparations, respectively. (**F**) Analysis of pluripotency-related genes (**F**) and lineage-specific genes (**G**) expression in Amicqui-1 maintained on iMEF feeder layer (AMIQ hESC-iMEF) by reverse transcription qPCR (RT-qPCR); Amicqui-1 on hAEC feeder layer (AMIQ hESC-hAEC); feeder-free Amicqui-1 with conditioned media of iMEF (AMIQ hESC-iMEF-CM); feeder-free Amicqui-1 with conditioned media of hAEC (AMIQ hESC-

*Figure 1 continued on next page*

*Figure 1 continued*

hAEC-CM). Data are mean ± SE, n=three biological samples per group, and three repetitions per sample. * p<0.05, ** p<0.01, and **** p<0.0001 with AMIQ hESC-iMEF as control.

The online version of this article includes the following source data and figure supplement(s) for figure 1:

**Source data 1.** Raw unedited blots (replicate one) for detection of NANOG and OCT4 in human embryonic stem cells (hESCs).

**Source data 2.** Raw unedited blots (replicate two) for detection of OCT4 and NANOG in human embryonic stem cells (hESCs).

**Source data 3.** Raw unedited blots (replicate three) for detection of NANOG and OCT4 in human embryonic stem cells (hESCs).

**Source data 4.** Uncropped blots with the bands labeled of representative western blot *Figure 1D*.

**Figure supplement 1.** Detection of pluripotency markers in H1 human embryonic stem cell (hESC) on inactivated mouse embryonic fibroblast (iMEF) or human amniotic epithelial cell (hAEC).

**Figure supplement 2.** Detection of pluripotency markers in H9 human embryonic stem cell (hESC) on inactivated mouse embryonic fibroblast (iMEF) or human amniotic epithelial cell (hAEC).

**Figure supplement 3.** Characterization of human embryonic stem cell (hESC) lines maintained on feeder-free conditions.

**Figure supplement 4.** Comparison of AMIQ human embryonic stem cell (hESC)-human amniotic epithelial cell (hAEC) between primed and naïve conditions in the expression of pluripotency-related genes and lineage-specific genes.

---

and H9), indicating that gene expression results were exclusively from embryonic stem cells and not a mixture of both populations (data not shown).

Pluripotency core expression was not statistically different between AMIQ hESC-hAEC and conventional AMIQ hESC-iMEF determined by quantitative PCR (qPCR). The hAEC alone had a POU5F1 level similar to both lines, but NANOG and SOX2 expression were almost undetectable (*Figure 1B*). We also did not find qualitative differences in the OCT4, SOX2 and NANOG (OSN) 'core' or other pluripotent-related proteins, such as TRA-1–60 and E-cadherin, by immunofluorescence (*Figure 1C*).

Afterward, to demonstrate that the hAEC feeder layer supports other hESC lines, H1 and H9 were maintained on hAEC for 10 consecutive passages. The cells constituted uniform and compact colonies, preserving the morphology growing on iMEF previously reported. We did not find any differences in the pluripotency markers through immunofluorescence and qPCR between the H1 and H9 colonies cultured on iMEF or hAEC (*Figure 1—figure supplement 1* for H1 and *Figure 1—figure supplement 2* for H9). Also, no differences were found in OCT4 and NANOG protein levels determined by western blot between AMIQ hESC-hAEC and H1 and AMIQ hESC-iMEF (*Figure 1D*). These results suggest that there are no differences in the hESC 'pluripotency core' regardless of whether they grow on an alternative hAEC or conventional iMEF feeder layer.

Next, we investigated whether the hAEC feeder layer can maintain hESC without cell-to-cell interaction. hESC lines (Amicqui-1, H1, and H9) were cultured on Matrigel with hAEC conditioned media (hAEC-CM) or iMEF conditioned media (iMEF-CM) during 10 passages. In both feeder-free conditions, the cells changed their morphology, growing in a monolayer until they reached confluence, as reported in the literature (*Figure 1—figure supplement 3*). The qPCR analysis showed no differences in the relative expression of hESC-hAEC-CM versus hESC-iMEF-CM, whereas the 'pluripotency core' immunodetection was similar between both conditions for the three evaluated lines (*Figure 1—figure supplement 3*). Finally, a high-throughput classification method (Epi-Pluri-Score *Lenz et al., 2015*) was performed to validate the previous profile characterization. The analysis determined that hESC-hAEC and hESC-iMEF and feeder-free hESC cultures (hESC-hAEC-CM and hESC-iMEF-CM) classified as pluripotent, while hAEC alone did not (*Figure 1E*). These data suggest that the hAEC feeder layer and its secretome (hAEC-CM) can support the undifferentiated state of hESC.

## The hESC-hAEC condition enhances the anterior neuroectoderm marker (OTX2) expression but not the mesendoderm lineage-specific genes

Since OSN expression profile is analogous between hESC-hAEC and conventional hESC, we determined differences of pluripotency-associated and lineage-specific genes between several AMIQ hESC

culture conditions (hESC-hAEC, hESC-iMEF, hESC-hAEC-CM, and hESC-iMEF-CM). The qPCR analysis showed no differences for ZFP42, TBX3, PRDM14, and KLF4 in the Amicqui-1 line under the four groups (*Figure 1F*), suggesting that the pluripotent global network remains unaltered regardless of the experimental environment. Then, we examined differences in lineage-specific gene expression (SOX17 and GATA6 for endoderm, T for mesoderm, and OTX2 for ectoderm). AMIQ hESC-hAEC, AMIQ hESC-iMEF-CM, and AMIQ hESC-hAEC-CM decreased SOX17 and GATA6 expression ($p<0.05$ and $p<0.01$; $p<0.05$ and $p<0.01$; $p<0.01$ and $p<0.01$, respectively) as compared with AMIQ hESC-iMEF. For the mesoderm lineage, we did not find differences for T (Brachyury) in any condition. Interestingly, OTX2 expression increased only in the AMIQ hESC-hAEC condition ($p<0.0001$) (*Figure 1G*). This transcription factor is vital during mESC to EpiSC transition and anterior neural fate specification. It is also a lineage-specific marker in primed hPSC (*Acampora et al., 2013*; *Buecker et al., 2014*; *Liu et al., 2017*).

## The hESC-hAEC condition does not express genes related to naïve pluripotency

We investigated if the AMIQ hESC-hAEC condition expresses genes related to naïve pluripotency. Amicqui-1 was cultured according to two protocols for its conversion into the naïve state (*Duggal et al., 2015*; *Zimmerlin et al., 2016*). We then compared its profile with that of the AMIQ hESC-hAEC condition. As expected, since the first passage, AMIQ hESC changed their morphology to dome-shaped colonies with smaller cells in both naïve protocols (*Figure 1—figure supplement 4A*). POU5F1 ($p<0.01$ and $p<0.001$ versus Duggal and Zimmerlin conditions, respectively) and SOX2 ($p<0.0001$ versus both conditions) expression levels increased in both naïve conditions as compared with AMIQ hESC-iMEF and AMIQ hESC-hAEC conditions. However, we did not find differences in NANOG expression in any of the conditions examined by qPCR (*Figure 1—figure supplement 4B*). The expression of naïve markers, such as STELLA ($p<0.0001$), KLF4 ($p<0.0001$), DPPA5 ($p<0.0001$), GBX2 ($p<0.05$), and LBP9 ($p<0.0001$), as well as human epiblast markers GDF3 ($p<0.0001$), KLF17 ($p<0.0001$), and TFAP2C ($p<0.05$), was upregulated only in the Zimmerlin condition, while only the expression level of GBX2 increased ($p<0.01$) in the Duggal condition (*Figure 1—figure supplement 4C-D*). Interestingly, the lineage-specific gene expression related to primed pluripotency (GATA6, SOX17, OTX2, and T) was not downregulated under both naïve conditions; even the mesoderm marker T increased compared to conventional hESC-iMEF (*Figure 1—figure supplement 4E*). Moreover, although the AMIQ hESC-hAEC condition does not have the gene profile associated with the naïve state, it exhibited the lowest expression levels of endoderm markers (GATA6 and SOX17), and OTX2 exhibited the highest expression as compared with its naïve counterparts (*Figure 1—figure supplement 4E*). These data indicate that AMIQ hESC-hAEC is not a culture condition that simulates the naïve pluripotent state in vitro. Its low lineage-specific gene expression is not analogous to the conventional primed hESC-iMEF molecular profile, suggesting that the hESC-hAEC condition may have a different molecular network representing a pluripotent identity other than naïve or primed.

## The hESC-hAEC condition displays a divergence in signaling pathways in contrast with the conventional culture

FGF2 is the main factor required to maintain undifferentiated hESC (*Eiselleova et al., 2009*; *Ludwig et al., 2006*; *Mossahebi-Mohammadi et al., 2020*). On the other hand, the human amnion secretes various growth factors, including the Fibroblast Growth Factor (FGF) family (*Grzywocz et al., 2014*; *Koizumi et al., 2000*), so we removed the exogenous FGF2 from the culture media to investigate whether the amnion-specific release factors were sufficient to maintain hESC. The absence of FGF2 for 48 hr induced the partial loss of OCT4 and NANOG in AMIQ hESC-hAEC, suggesting a differentiation process. In contrast, immunodetection of each pluripotent marker was not reduced in AMIQ hESC-iMEF (*Figure 2—figure supplement 1*). These data suggest that the FGF2 pathway is fundamental to maintain the AMIQ hESC pluripotency on hAEC. Also, the quick disappearance of the pluripotency factors signal in hESC-hAEC in the absence of exogenous FGF2 involves different signaling pathways that could trigger differentiation in our alternative condition. To prove it, we determined the protein phospho-kinase and cytokine levels throughout proteome arrays in Amicqui-1 with the hAEC and iMEF co-culture, H1 with iMEF (control), and hAEC alone. The AMIQ hESC-hAEC condition showed the highest levels of phosphorylation to EGFR, AMPKα1/α2, mTOR, AKT1S1, HSPB1, AKT1/2/3-S473,

and CHK2 (*Figure 2A*) as well as several members of the SCR kinase family (*Figure 2B*) as compared with the rest of the groups.

We detected a higher level of phospho-kinases, such as p53 (S392 and S46), p70 (S6-T389 and S6-T421/S424), STAT3 (Y705 and S727), SK1/2/3, and ERK1/2, in AMIQ hESC-hAEC and hAEC alone compared with H1 hESC-iMEF (*Figure 2C*). Regarding cytokines and growth factors released to the culture media, the AMIQ hESC-hAEC condition showed higher levels of HGF, IL1B, IL11, IL6, IL19, IL10, and PDGF-AB/BB than H1 and AMIQ hESC-iMEF as well as hAEC (*Figure 2D*). We also detected a higher expression of cytokines (LCN2, CD40LG, DKK1, IGFBP3, IL1RN, CCL19, MMP9, CCL17, and THBS) in the AMIQ hESC-hAEC medium and in hAEC alone, as compared with the hESC-iMEF condition (*Figure 2E*). Afterward, we used the STRING database enrichment analysis to determine functional interactions between phospho-kinases and cytokines in the AMIQ hESC-hAEC condition and found that our set of proteins are partially connected (p<1.0e−16) (*Figure 2—figure supplement 2*). Analysis with Gene ontology (GO) indicated that they are involved in several biological processes, such as 'regulation of protein metabolic process' (p=9.61e−18), 'negative regulation of apoptosis' (p=3.99e−17), 'localization' (p=7.69e−13), 'regulation of developmental process' (p=5.36e−12), 'transcription by RNA polymerase II' (p=2.79e−06), 'RNA metabolic process' (p=7.86e−06), and 'regulation of multicellular organism development' (p=1.18e−10) (*Figure 2—figure supplement 3*).

Further, cytokines and phospho-kinases participated in cytokine-cytokine receptor interaction and PI3K-AKT, mTOR, MAPK (ERK1/2), JAK-STAT, and insulin pathways throughout KEGG and Reactome analysis pathways (*Figure 2F*).

Notably, PI3K/AKT/mTOR, MAPK, and JAK/STAT pathways, considered as part of the pluripotent identity, are more enriched in AMIQ hESC-hAEC than in conventional AMIQ hESC-iMEF. We corroborated the presence of one representative component of each path (AKT1, ERK1/2, and STAT3, respectively) by immunodetection (*Figure 2G*), suggesting a confluence of routes in the AMIQ hESC-hAEC condition.

On the other hand, the SRC kinase family, which induces the hESC differentiation process (*Zhang et al., 2014*), was found in higher levels in AMIQ hESC-hAEC compared with the remaining conditions (*Figure 2B*). When FGF2 was removed from the culture medium, the loss of pluripotency factors in AMIQ hESC-hAEC was faster than in hESC-iMEFs. Hence, it is likely that SRC pathways participate in differentiation processes in the AMIQ hESC-hAEC condition once the maintenance of exogenous FGF2-dependent pluripotency is disrupted. All these data indicate that the hESC-hAEC co-culture promotes the differential activation of signaling pathways with respect to the conventional hESC-iMEF condition, which would confer it an alternative molecular identity.

## hESC-hAEC exhibits a unique transcriptome signature different from that of conventional hESC and other pluripotency conditions

To determine if the transcription profile of AMIQ hESC-hAEC differs from that of H1 and AMIQ hESC-iMEF, we carried out RNA-seq on both feeder layer conditions and on their respective conditioned media (hESC-hAEC-CM and hESC-iMEF-CM) and hAEC alone. Differential gene expression (DEG) analysis using the Limma package (logFC >1.5, p-value<0.05) indicated 2550, 2537, 1105, 865, and 5303 genes differentially expressed in AMIQ hESC-hAEC versus H1 hESC-iMEF, AMIQ hESC-iMEF, AMIQ hESC-iMEF-CM, AMIQ hESC-hAEC-CM, and hAEC, respectively (*Figure 3—figure supplement 1*, *Supplementary file 1*). We also compared our raw value with the data of lines under alternative conditions, such as feeder-free H9 hESC-TesR1 (*Shao et al., 2017*), 3iL H1 hESC (*Chan et al., 2013*), naïve hESC [2iL +Gö HNES hESC (*Guo et al., 2017*) and 5iL/FA H9 hESC (*Collier et al., 2017*)], conventional H9 hESC-iMEF (*Collier et al., 2017*), in formative transition [HNES hESC with XAV939 (*Rostovskaya et al., 2019*), FTW1 and FTW2 hESCs (*Yu, 2021*), expanded hPSC (*Gao, 2019*) and hPSC-derived amnion (*Shao et al., 2017*; *Figure 3—figure supplement 1*, *Supplementary file 1*)].

Principal component analysis (PCA) showed AMIQ hESC-hAEC clustered with -iMEF (H1 hESC-iMEF, H9 hESC-iMEF, and AMIQ hESC-iMEF), -conditioned medium (CM)(AMIQ hESC-iMEF-CM and AMIQ hESC-hAEC-CM), expanded hPSC, FTW-hESC, H9 hESC-TesR1, and 3iL H1 hESC, while the naïve cells (2iL +Gö HNES hESC and 5iL/FA H9 hESC), XAV939 HNES1 and hAEC were found in a specific different cluster (*Figure 3A*). Hierarchical clustering of the transcriptome data situates AMIQ hESC-hAEC with its CM counterpart (AMIQ hESC-hAEC-CM). Intriguingly, AMIQ hESC-hAEC showed a higher correlation with H9 hESC-TesR1 (*Figure 3B*).

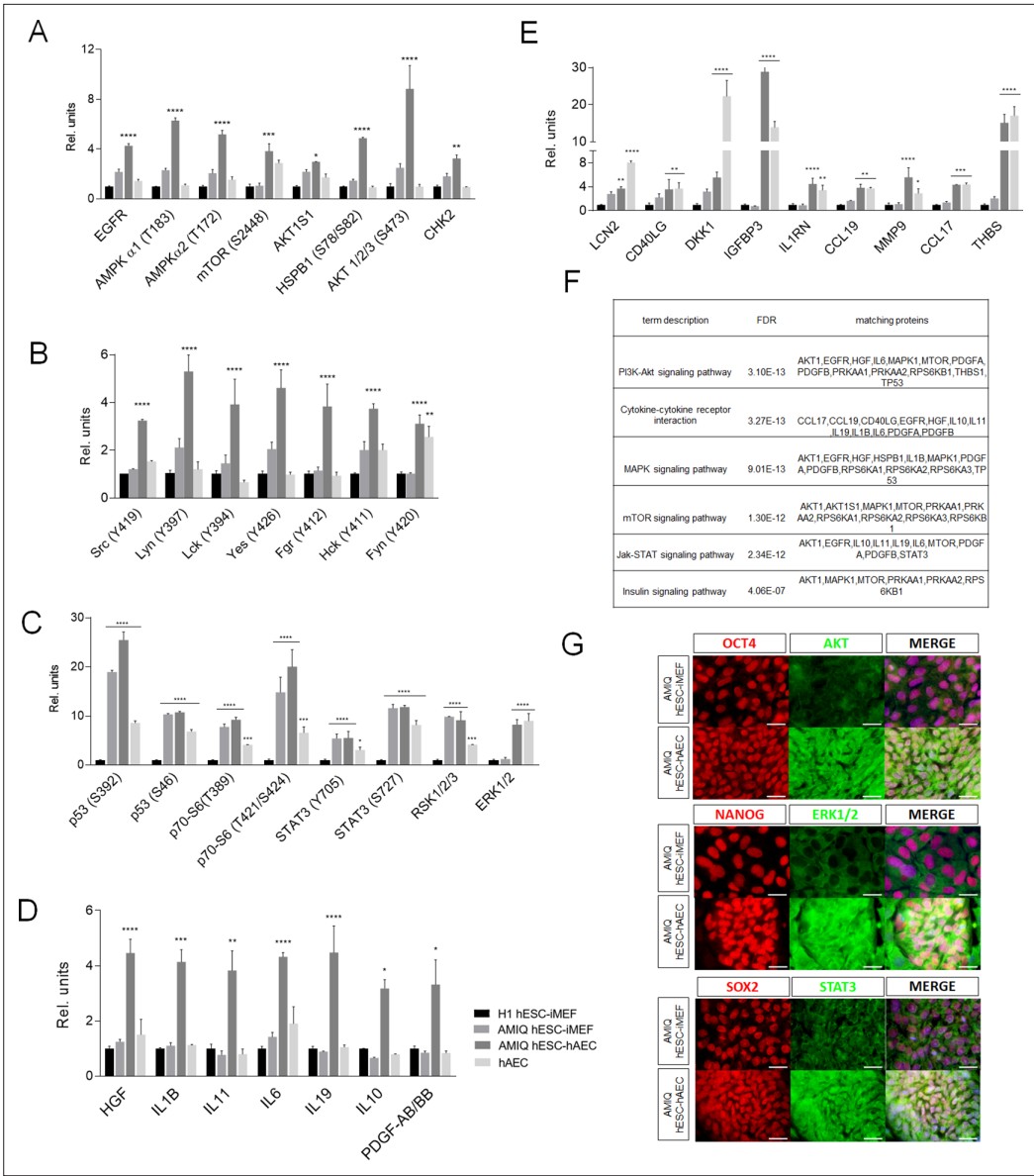

**Figure 2.** Different signaling pathways related to pluripotency and development converge in AMIQ human embryonic stem cell (hESC)-human amniotic epithelial cell (hAEC). (**A**) Phospho-kinases levels increased in Amicqui-1 hESC maintained on hAEC feeder layer (AMIQ hESC-hAEC). (**B**) Phospho-kinases levels of SRC family in AMIQ hESC-hAEC. (**C**) Phospho-kinases levels increased in both AMIQ hESC-hAEC and hAEC. (**D**) Cytokines levels increased in AMIQ hESC-hAEC. (**E**) Cytokines levels increased in both AMIQ hESC-hAEC and hAEC. * p<0.05, ** p<0.01, ***p<0.001, and **** p<0.0001 using H1 hESC maintained on inactivated mouse embryonic fibroblast (iMEF) as control (H1 hESC-iMEF). For each phospho-kinase and cytokine detected, data are mean ± SE, n=three biological samples per group, and two repetitions per sample. (**F**) List of selected Kyoto Encyclopedia of Genes and Genomes (KEGG) and Reactome pathways obtained from Gene ontology (GO) analysis based on the proteins identified in proteome arrays. (**G**) Presence of AKT, ERK1/2, and STAT3 in AMIQ hESC-hAEC. Representative epifluorescence micrographs of double immunostaining for OCT4 with AKT, NANOG with ERK1/2, and SOX2 with STAT3 in AMIQ hESC-iMEF and AMIQ hESC-hAEC. All the images were obtained with an epifluorescence microscope with the same gain and exposure parameters. Scale bar, 50 μm.

The online version of this article includes the following figure supplement(s) for figure 2:

**Figure supplement 1.** AMIQ human embryonic stem cell (hESC)-human amniotic epithelial cell (hAEC) depends on FGF2 exogenous to maintain pluripotency-related transcription factors.

*Figure 2 continued on next page*

*Figure 2 continued*

**Figure supplement 2.** Known and predictive interactions of cytokines and kinases present in AMIQ human embryonic stem cell (hESC)-human amniotic epithelial cell (hAEC).

**Figure supplement 3.** Gene ontology (GO) term analyses for biological processes based on the proteins identified in the proteome arrays.

---

To identify if the AMIQ hESC-hAEC condition corresponded to a specific lineage in the human embryo, we carried out a computational deconvolution analysis (CIBERSORT *Newman et al., 2015*) to estimate the relative similarity between the bulk RNA-seq data of the cell lines and the scRNA-seq obtained from human epiblast and pre-gastrula (*Tyser, 2021*; *Xiang et al., 2020*). As expected, the naïve hPSC lines had a remarkable similarity with the pre-implantation epiblast (E7). In contrast, the remnant of the groups had a higher score with the post-implantation epiblast E14, including AMIQ hESC-hAEC (*Figure 3C*). hPSC lines correspond to the epiblast as compared with human pre-gastrula lineages, as expected. However, the hAEC were related to the extraembryonic ectoderm, corresponding to the nascent amnion, although also to the extraembryonic mesoderm (*Figure 3C*). Conventional primed conditions had a minimum score (0.16–0.30) with the primitive streak or emergent mesoderm (0.16–0.39), naïve conditions with the emergent mesoderm (0.21–0.22), while formative hiPSC (FTW) and expanded hPSCs shared a score with the advanced mesoderm (0.23–0.24) and nascent mesoderm (0.51 for FTW hiPSC). Interestingly, AMIQ hESC-hAEC together with TesR1 H9 and XAV939 HNES1 showed no relationship with embryonic and extraembryonic mesodermal lineages (*Figure 3C*).

A KEGG-based analysis was carried out to identify biological pathways overrepresented (upregulated genes) in AMIQ hESC-hAEC compared to the rest of the conditions (*Supplementary file 2*). We detected an enrichment in the Hippo, Hedgehog, extracellular matrix (ECM)-receptor interaction, Focal adhesion, PI3K-Akt, and Wnt signaling pathways (*Figure 3—figure supplement 2*). The DEG analysis demonstrated that genes involved in these pathways, such as IGF2 and IGBP3 (for PI3K/Akt path), LAMA3 and LAMB3 (ECM-receptor interaction), increase their expression in AMIQ hESC-hAEC, unlike in all other groups. Intriguingly, concerning the Wnt pathway, WNT6 (involved in neural crest induction *Schmidt et al., 2007*) was upregulated, while WNT3 (necessary for primitive streak establishment *Yoon et al., 2015*) was down-regulated in AMIQ hESC-hAEC (*Figure 3—figure supplement 3*), supporting the low similarity score with the primitive streak from the computational deconvolution analysis (*Figure 3C*). Indeed, ECM-receptor interaction and Insulin-like Growth Factor (IGF) and WNT pathways have been suggested as markers to evaluate the interaction between epiblast and polar trophectoderm in human embryo (*Meistermann et al., 2021*).

Next, we compared the biological processes and reactome from AMIQ hESC-hAEC versus the different culture hESC conditions using enrichment pathways (p<0.01) (*Reimand et al., 2019*). We found several similar pathway clusters related to the regulation of morphogenesis (animal organ morphogenesis, anatomical structure morphogenesis, tissue morphogenesis, animal organ development, etc.) and other smaller clusters involved in cell adhesion, type I hemidesmosome and cell-junction assemblies and ECM organization upregulated in AMIQ hESC-hAEC versus H1 hESC-iMEF, H9 hESC-iMEF, and AMIQ hESC-iMEF (*Figure 3—figure supplement 4A*). Comparison with the TesR1 condition showed upregulation of morphogenesis processes, type I hemidesmosome assembly and cell adhesion, just like the comparison with H1 hESC-iMEF. Also, glucose metabolism, reactive oxygen species, apoptosis, and Wnt and IGF pathways were upregulated (*Figure 3—figure supplement 4B*). When we compared AMIQ hESC-hAEC and the AMIQ hESC-iMEF-CM conditions, we found a similar profile against H1 hESC-iMEF and TesR1 conditions (upregulation of morphogenesis, type I hemidesmosome and cell-junction assemblies, ECM organization, and Wnt and IGF pathways) (*Figure 3—figure supplement 4C*). Interestingly, compared to AMIQ hESC-hAEC-CM, only the hemidesmosome assembly and animal organ development (forebrain development and regionalization) were upregulated (*Figure 3—figure supplement 4D*). When we compared AMIQ hESC-hAEC with naïve conditions (2iL +Gö and 5iL/A), we found that morphogenesis, cell junction, cell adhesion, and ECM organization, like under the other conditions, were upregulated. Additionally, 'defective ext 2' and 'synaptic signaling' were upregulated (*Figure 3—figure supplement 4E*). The comparison between AMIQ hESC-hAEC and the 3iL naïve condition showed similar differences against H1 hESC-iMEF and TesR1 (regulation of morphogenesis, hemidesmosome assembly, ECM organization, and Wnt and IGF

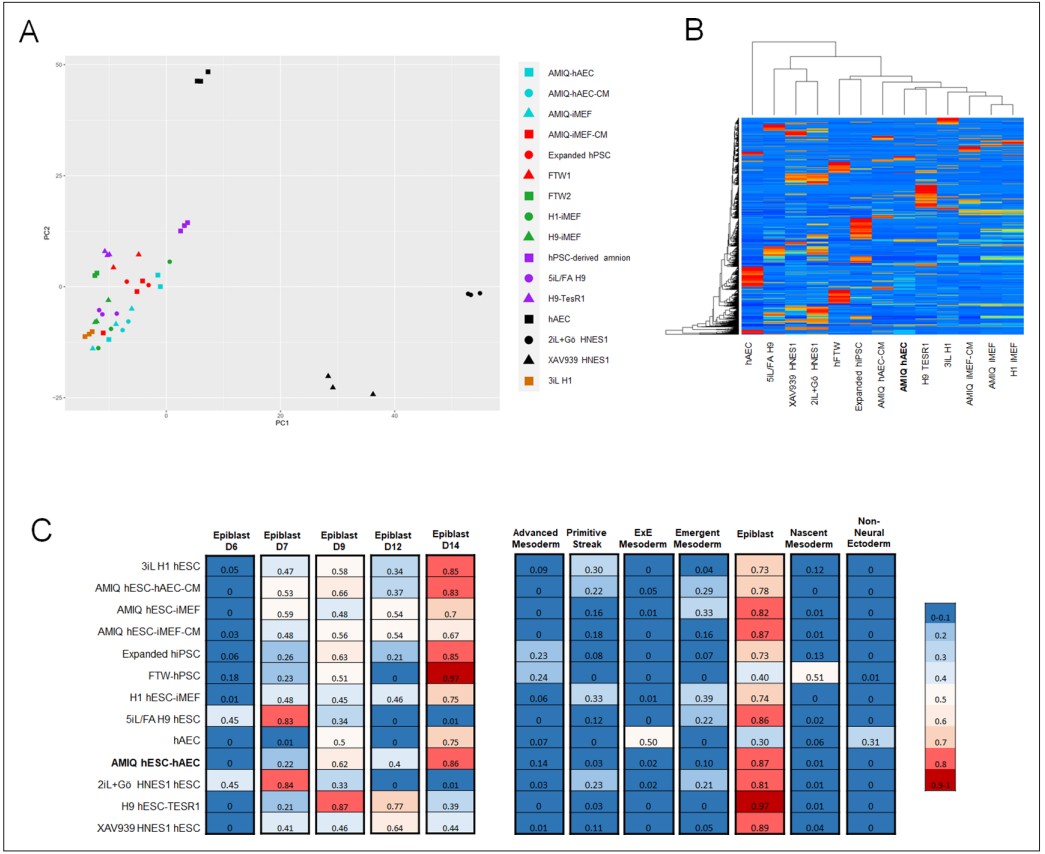

**Figure 3.** Comparison of the genome-wide expression of AMIQ human embryonic stem cell (hESC)-human amniotic epithelial cell (hAEC) with other human pluripotent stem cell (hPSC) lines and human embryonic lineages. (**A**) Principal component analysis (PCA) of bulk RNA-seq data of hPSC lines. AMIQ hAEC, Amiqui hESC on alternative hAEC feeder layer; AMIQ hAEC-CM, feeder-free Amicqui-1 hESC with hAEC-conditioned media; AMIQ hESC-inactivated mouse embryonic fibroblast (iMEF), Amicqui-1 hESC on conventional iMEF feeder layer; AMIQ hESC-iMEF-CM, feeder-free Amicqui-1 hESC with iMEF-conditioned media; expanded hPSC, expanded human induced PSC (hiPSC); FTW1 and FTW2, formative hiPSCs; H1 iMEF, H1 hESC on conventional iMEF feeder layer; H9 hESC-iMEF, H9 hESC on conventional iMEF; hPSC-derived amnion, amnion derived from H9 hESC; 5iL/FA H9, naïve 5iL/FA H9 hESC; H9 hESC-TesR1, feeder-free H9 hESC with TesR1 medium; hAEC, hAEC without co-culture with hESC; 2iL +Go HNES1, naïve 2iL +Go HNES1 hESC; XAV939 HNES1, formative XAV939 HNES1 hESC; 3iL H1, 3iL H1 hESC. (**B**) Heatmap of signature matrix for differential expression genes of hPSC lines. (**C**) Relative similarity of the bulk RNA-seq of the hPSC lines with the single cell RNA sequencing (scRNA-seq) of human embryos.

The online version of this article includes the following figure supplement(s) for figure 3:

**Figure supplement 1.** The number of genes differentially expressed in AMIQ human embryonic stem cell (hESC)-human amniotic epithelial cell (hAEC) versus human pluripotent stem cell (hPSC) lines.

**Figure supplement 2.** KEGG pathways upregulated in AMIQ human embryonic stem cell (hESC)-human amniotic epithelial cell (hAEC) versus human pluripotent stem cell (hPSC) lines.

**Figure supplement 3.** LogFC from differential gene expression (DEG) analysis (FC<1.5 and p-value<0.05) for genes in AMIQ human embryonic stem cell (hESC)-human amniotic epithelial cell (hAEC) compared with conventional primed, feeder-free, naïve, expanded, and XAV939 conditions as well as versus H9-derived amnion and hAEC isolated from membranes at term.

**Figure supplement 4.** Enrichment maps of biological processes that differ in AMIQ human embryonic stem cell (hESC)-human amniotic epithelial cell (hAEC) versus several pluripotency conditions.

**Figure supplement 5.** Enrichment maps of biological processes differ in AMIQ human embryonic stem cell (hESC)-human amniotic epithelial cell (hAEC) versus amnion-like cells derived from H9 (**A**) and hAEC isolated from amnion at term (**B**).

**Figure supplement 6.** Differential expression of HOX genes in AMIQ human embryonic stem cell (hESC)-human amniotic epithelial cell (hAEC).

signaling pathways) and leukocyte migration (*Figure 3—figure supplement 4E*). AMIQ hESC-hAEC presented only a specific downregulation of processes grouped in 'response to metal ions' compared with the three naïve conditions (2iL +Gö, 5iL/A, and 3iL) (*Figure 3—figure supplement 4E*).

Although expanded cells are different because of their pluripotent potential and differentiation toward extraembryonic lineages, the only upward process that differentiates AMIQ hESC-hAEC from expanded cells is the regulation of hemidesmosome and cell junction assemblies, cell adhesion, and epithelium development (*Figure 3—figure supplement 4F*).

Cell adhesion, hemidesmosome assembly, ECM organization, and IGF pathways were also upregulated against the XAV939 hESC condition. The Wnt pathway was upregulated as expected (XAV939 is an inhibitor of Wnt signaling), but biological processes involved in neural development (nervous system regulation, axogenesis, and forebrain development) predominated in AMIQ hESC-hAEC versus this condition (*Figure 3—figure supplement 4G*). The comparison between transcriptome data indicated that the molecular identity of AMIQ hESC-hAEC differs from that of the other hESC/hPSC maintained in conventional, naïve, expanded, and capacitance conditions.

IGF signaling was a predominant pathway in AMIQ hESC-hAEC compared with the feeder-free hESC culture (TesR1 and AMIQ hESC-iMEF-CM), 3iL, and XAV939 hESC. This signaling (via PI3K/mTOR) with Activin supports hESC pluripotency in the absence of FGF2 (*Wamaitha et al., 2020*). Although our enrichment analysis did not find differences for this pathway between AMIQ hESC-hAEC and H1 hESC-iMEF, we detected a higher release of IGFBP3 into the conditioned media through the proteome arrays exclusively in AMIQ hESC-hAEC and hAEC compared with H1 hESC-iMEF (*Figure 2E*). In harmony with these data, DEG analysis indicated that IGFBP3 and IGF2 were upregulated in AMIQ hESC-hAEC compared with all conditions (*Figure 3—figure supplement 3*). However, both genes were downregulated in AMIQ hESC-hAEC as compared with hAEC and the hPSC-derived amnion. Recently, single-cell characterization of the human embryonic gastrula identified IGFBP3 as part of the molecular identity of the amniotic ectoderm (*Tyser, 2021*). These data could be suggesting a presumptive role of the IGF pathway in the interaction of embryonic cells and the nascent amnion.

## Expression of 'formative' and 'intermediate' genes in hESC-hAEC

It has been postulated that formative pluripotency is an intermediate phase between the naïve and primed states in mESC (*Kalkan et al., 2017*). However, there is no confirmation of their in vitro capture in humans. When we compared H1 and H9 hESC-iMEF, AMIQ hESC-iMEF-CM and AMIQ hESC-hAEC-CM versus AMIQ hESC-hAEC, the transcription factors SOX3 (associated with the formative stage in mESC) were not overexpressed in any condition. Nonetheless, POU3F1, SALL2, and OTX1 increased according to the DEG analysis. This result was validated by the qPCR array in AMIQ hESC-hAEC (*Figure 4A*).

In the formative pluripotency stage, the cells could acquire competence to specify toward the germline (*Smith, 2017*). To demonstrate whether the co-culture of hESC-hAEC acquires functional properties of the formative pluripotency and not only POU3F1, OTX1, and SALL2 expression, we carried out a differentiation protocol toward human primordial germ cells (hPGC) in AMIQ hESC-hAEC (*Figure 4—figure supplement 1A*), AMIQ hESC-iMEF (*Figure 4—figure supplement 2*), and H9 hESC-iMEF (*Figure 4—figure supplement 3*) and compared their efficiency between groups (*Figure 4—figure supplement 1B*). hESCs were detached from their respective feeder layer and cultured on Matrigel under mesoderm-like cell (MeLC) induction conditions for 2 days. Subsequently, the cells were subjected to Bone Morphogenetic Protein 4 (BMP4), Epidermal Growth Factor (EGF), Leukemia Inhibitory Factor (LIF) and Stem Cell Factor (SCF) treatment. On day 9 (D9) and day 12 (D12), after initiating the differentiation protocol, we evaluated the TFAP2C, BLIMP1, and STELLA germline markers. At D9, we observed a higher number of TFAP2C/STELLA double-positive cells in AMIQ hESC-hAEC compared with MEF conditions; it was also the only condition with BLIMP1/STELLA double-positive cells. Although all groups presented the three germline markers at D12, we did not observe the BLIMP1+/STELLA+ colocalization in the hESC-iMEF groups (*Figure 4—figure supplement 1B*). These results suggested that hESC on hAEC have a better predisposition to differentiate toward the germline.

During the formative transition, cells acquired an epithelial phenotype and increased their interaction with the ECM, events displayed during the implantation period (*Smith, 2017*). Interestingly, the GO analysis showed an increase in 'ECM organization,' 'cell and biological adhesion,' and 'epithelium

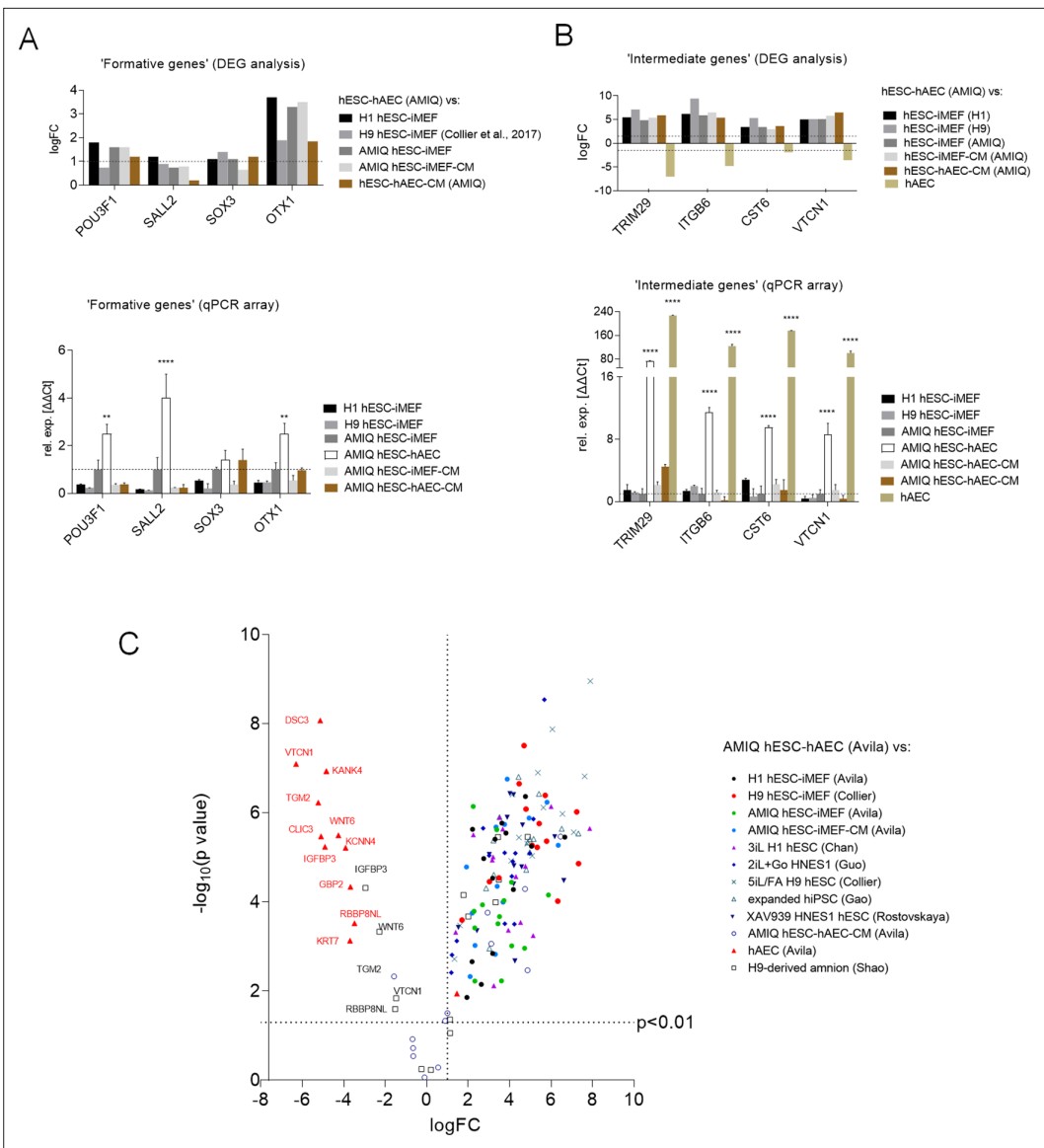

**Figure 4.** Expression of formative pluripotency-associated genes and 'intermediate genes' in AMIQ human embryonic stem cell (hESC)-human amniotic epithelial cell (hAEC). (**A**) Differential gene expression (DEG) analysis and validation by a quantitative PCR (qPCR) array of 'formative genes' upregulated in Amicqui-1 hESC maintained on hAEC feeder layer (AMIQ hESC-hAEC). Data are mean ± SE, n=three biological samples per group, and two repetitions per sample. ****$p<0.0001$ using Amicqui-1 hESC on inactivated mouse embryonic fibroblast (iMEF) feeder layer (AMIQ hESC-iMEF) as a control for qPCR. FC<1.0 and $p<0.05$ for DEG. (**B**) DEG analysis and validation by a qPCR array of 'intermediate genes' upregulated in AMIQ hESC-hAEC. Data are mean ± SE, n=three biological samples per group, and two repetitions per sample. ****$p<0.0001$ using AMIQ hESC-iMEF control for qPCR. FC<1.0 and $p<0.05$ for DEG. (**C**) AMIQ hESC-hAEC upregulation of genes associated with an 'intermediate population'. List of genes: *C1orf21*, *CERCAM*, *DSC3*, *CLIC3*, *GBP2*, *KCNN4*, *RBBP8NL*, *WNT6*, *IGFBP3*, *KANK4*, *KRT7*, *ODAM*, *TGM2*, and *VTCN1*. FC<1.5 and $p<0.05$. The conditions compared to AMIQ hESC-hAEC were: H1, H9, and Amicqui-1 hESC on iMEF feeder layer; feeder-free H9 hESC with TesR1; hESC on naïve conditions (3iL H1, 2iL +Go HNES1, and 5iL/FA H9); expanded hPSC; and hESC in formative transition (HNES1 line with XAV939 inhibitor).

The online version of this article includes the following figure supplement(s) for figure 4:

**Figure supplement 1.** Human embryonic stem cell (hESC)-human amniotic epithelial cell (hAEC) is competent to differentiate into germline-like cells.

*Figure 4 continued on next page*

*Figure 4 continued*

**Figure supplement 2.** Representative immunodetection micrographs of Amicqui-1 human embryonic stem cell (hESC) on inactivated mouse embryonic fibroblast (iMEF)-derived TFAP2C+, BLIMP1+, and STELLA+ cells on day 9 and day 12.

**Figure supplement 3.** Representative immunodetection micrographs of H9 human embryonic stem cell (hESC) on inactivated mouse embryonic fibroblast (iMEF)-derived TFAP2C+, BLIMP1+, and STELLA+ cells on day 9 and day 12.

**Figure supplement 4.** Biological processes involved in epithelialization and extracellular matrix organization upregulated in AMIQ human embryonic stem cell (hESC)-human amniotic epithelial cell (hAEC).

development' in AMIQ hESC-hAEC compared with the other conditions (*Figure 4—figure supplement 4*), following the KEGG results (ECM-receptor interaction pathway) (*Figure 3—figure supplement 2*). Furthermore, DEG analysis showed an upregulation of epithelialization and interactions with ECM genes in the AMIQ hESC-hAEC condition. Hence, we determined TRIM29, ITGB6, CST6, and VTCN1 expression. These genes were selected because TRIM29 and CST6 are enriched in the squamous epithelium, while the ITGB6 (fibronectin receptor) signals from the ECM to the cells, according to the Human Protein Atlas (*Uhlen, 2015*). We found a higher expression in AMIQ hESC-hAEC as compared with the other groups (*Figure 4B*).

On the other hand, a recent study demonstrated the existence of an 'intermediate' subpopulation, which exhibited a specific gene signature involved in morphological structure formation and development in hESC (*Messmer et al., 2019*). We explored whether the AMIQ hESC-hAEC condition expresses these intermediate genes. Through DEG analysis, we found that CERCAM, DSC3, CLIC3, GBP2, KCNN4, IGFBP3, KANK4, C1orf21, KRT7, ODAM, WNT6, TGM2, RBBP8NL, and VTCN1, which correspond to the intermediate gene set, were upregulated in AMIQ hESC-hAEC (FC>1.5, p<0.05) as compared with primed (feeder and feeder-free conditions), naïve (2iL +Gö, 3iL, 5iL/FA and XAV939), and expanded hPSC (*Figure 4C* and *Supplementary file 3*). Interestingly, VTCN1, IGFBP3, CLIC3, C1orf21, KRT7, and WNT6 were associated with the amniotic ectoderm identity in human gastrulating embryo (*Tyser, 2021*). Indeed, these genes were expressed in cells exiting from the naïve state to continue with the formative pluripotency. Nonetheless, POU3F1, OTX2, and SOX3 expression (associated with formative pluripotency in mice) was not reported (*Messmer et al., 2019*). In contrast, we found that the AMIQ hESC-hAE condition expressed POU3F1, OTX1 (*Figure 4A*), and OTX2 (*Figure 1G*), and genes also enriched in the intermediate population. The 'intermediate' genes (except C1orf21) were downregulated in AMIQ hESC-hAEC compared to hAEC isolated from amnion at term; whereas the expression of IGFBP3, RBBP8NL, WNT6, and VTCN1 was higher in the hPSC-derived amnion than in AMIQ hESC-hAEC (*Figure 4C*).

VTCN1 and CERCAM are localized in cell junctions of epithelial cells, while DSC3 is distributed in squamous epithelia (amnion cells are considered 'squamous epithelia'), and ODAM can be detected in the epithelial cell nucleoplasm. Interestingly, the intermediate population of embryonic cells, not naïve or primed, is considered as formative because of their epithelization. It has also been demonstrated that the derivation of amnion-like cells from hESC involves 'intermediate' gene upregulation (*Shao et al., 2017*).

These data suggest that a subpopulation of pluripotent cells that express the intermediate genes could be predisposed to differentiate into the squamous epithelium or amniotic ectoderm. Thus, formative cells can differentiate into the nascent amnion instead of naïve or primed cells. As a first approach to investigate the processes involved in the transition of embryonic cells to the amnion, we compared the AMIQ hESC-hAEC condition with the hPSC-derived amnion and hAEC at term. We found an upregulation of morphogenesis, hemidesmosome assembly, and cell adhesion compared with the hPSC-derived amnion. Additionally, the degradation of extracellular collagen, glucose metabolism, phospholipase regulation, hydrolase, and endopeptidase activity predominated in these compared conditions (*Figure 3—figure supplement 5A*). When comparing AMIQ hESC-hAEC and hAEC isolated from the amnion, the predominant processes were involved in morphogenesis, regulation of pluripotent stem cells, protein localization, and synaptic processes including synaptic signaling, neurotransmitter receptor activity, and protein interactions at synapses (*Figure 3—figure supplement 5B*). Although AMIQ hESC-hAEC has a different transcriptomic profile from the other conditions, its identity is still pluripotent to differentiate itself from the amniotic epithelium.

## HOX genes are downregulated in the alternative hESC-hAEC condition

We found a downregulation in genes associated with 'DNA signaling' in the AMIQ hESC-hAEC group as compared with H1 hESC-iMEF, naïve 2iL +Gö hESC, and XAV939 hESC conditions. In this category, we included 'chromatin organization' and 'activation of HOX genes during differentiation' through enrichment analysis (*Figure 4A, D and F*). The DEG analysis confirmed HOX gene downregulation (FC<−1.0, p<0.05), particularly, in the central (HOX5-8) and posterior (HOX9-13) clusters in most conditions. However, when we compared with the TesR1 condition, there were no differences (*Figure 3—figure supplement 6*). In most bilaterian animals, HOX genes participate in anterior-posterior axis patterning during development. The spatially and temporally co-linear expression of HOX genes regulates the neural domains that pattern the most posterior structures of the neuroectoderm, hindbrain rhombomeres (HOX1-5), as well as spinal cord cervical (HOX5-9), thoracic (HOX9-10), and lumbosacral (HOX10-13) vertebral segments (*Philippidou and Dasen, 2013*). Together with a higher expression of OTX2, HOX gene downregulation suggested that AMIQ hESC-hAEC is predisposed to generate the anteriorized neuroectoderm (forebrain and midbrain) characterized by a lack of HOX gene expression.

## Anterior neural fate and differentiation into specific cortical layers are upregulated in hESC-hAEC

DEG analysis (FC>1.0 and p<0.05) revealed a set of neurodevelopmental regulatory genes (NRGs) upregulated in AMIQ hESC-hAEC in contrast with conventional primed, naïve, and expanded hPSC lines (*Figure 5A* and *Figure 5—figure supplement 1*).

Afterward, FEZF2, LHX5, RAX, and SIX3 were validated through the qPCR assay, and a higher expression was observed in AMIQ hESC-hAEC compared with H1, H9 and AMIQ hESC-iMEF, AMIQ hESC-iMEF-CM, and AMIQ hESC-hAEC-CM (*Figure 5B*). Interestingly, the gene assembles a predictive network with significant interaction enrichment (p-value<1.0e−16), indicating that the coding proteins are partially connected and involved in developing structures from the anteriorized neuroectoderm (*Figure 5—figure supplement 2*). Indeed, the role of these genes in neural stem and neural crest cells development (SOX9; *Jo et al., 2014*), formation of the anterior neural fold that gives rise to the ventral brain and optic vesicles (RAX; *Furukawa et al., 1997*), retina generation (SIX3, RAX, LHX2, and PAX; *Tétreault et al., 2009*), forebrain development (LHX2, LHX5, SIX3, and SOX21; *Peng and Westerfield, 2006*; *Peukert et al., 2011*), neural plate border formation (HES3 and OLIG3; *Filippi, 2005*; *Hong and Saint-Jeannet, 2018*), and neurogenesis control in dorsal telencephalon (FEZF1 and FEZF2; *Shimizu et al., 2010*) has been experimentally validated in vertebrates.

Since the forebrain is a primordial structure generated from the anterior neuroectoderm, we challenged the AMIQ hESC-hAEC through a neural induction protocol based on dual SMAD inhibition and compared their capacity to differentiate into cortical neurons versus H9 and AMIQ hESC-iMEF. At the end of neural induction (after 12 days of starting the protocol) and the proliferation (D18) stages, we detected the presence of neural rosettes (PAX6+/NESTIN+). Notwithstanding, we did not observe differences between the three culture conditions (*Figure 5—figure supplement 3*). From D25, we started detecting the first mature (MAP2+) neurons. Interestingly, the immunoreactivity to MAP2+ cells at D25 was higher in AMIQ hESC-hAEC as compared with AMIQ and H9 hESC-iMEF (*Figure 5C*). Deep layer cortical phenotypes (CTIP2 and FOXP2) were identified at D40, while at D50, GAD67+ and CALRETININ+ interneurons emerged. We found an increase in all the cortical markers evaluated in AMIQ hESC-hAEC compared with its analogs on iMEF cultures (*Figure 5D and E*). These data showed that the hESC co-culture with hAEC confers a superior functional potential to generate the neural lineage cells, specifically from the forebrain.

## Inhibition of SRC and ERK alters transcriptional identity and neural potential of hESC-hAEC

We suggested that NRG is transcriptionally repressed through the SUZ12 subunit of the polycomb repressive complex (PCR2) (*Lee et al., 2006*) in the conventional undifferentiated hESC-iMEF condition. The enrichment analysis of transcription factors (via Enrichr) indicates that several of upregulated genes in hESC-hAEC are targets of PCR1 (RNF2 /RINGB1, CBX2, PHC1, and BMI1) and PRC2 (EZH2, EED, SUZ12, and JARID) subunits (*Figure 5—figure supplement 4* and *Supplementary file 4*). Thus, we inferred that hAEC could secrete a molecule repressing the polycomb complex in hESC. Interestingly, the miR-200 family (miR200b/miR200c) directly inhibits the SUZ12 subunit of the PCR2 in

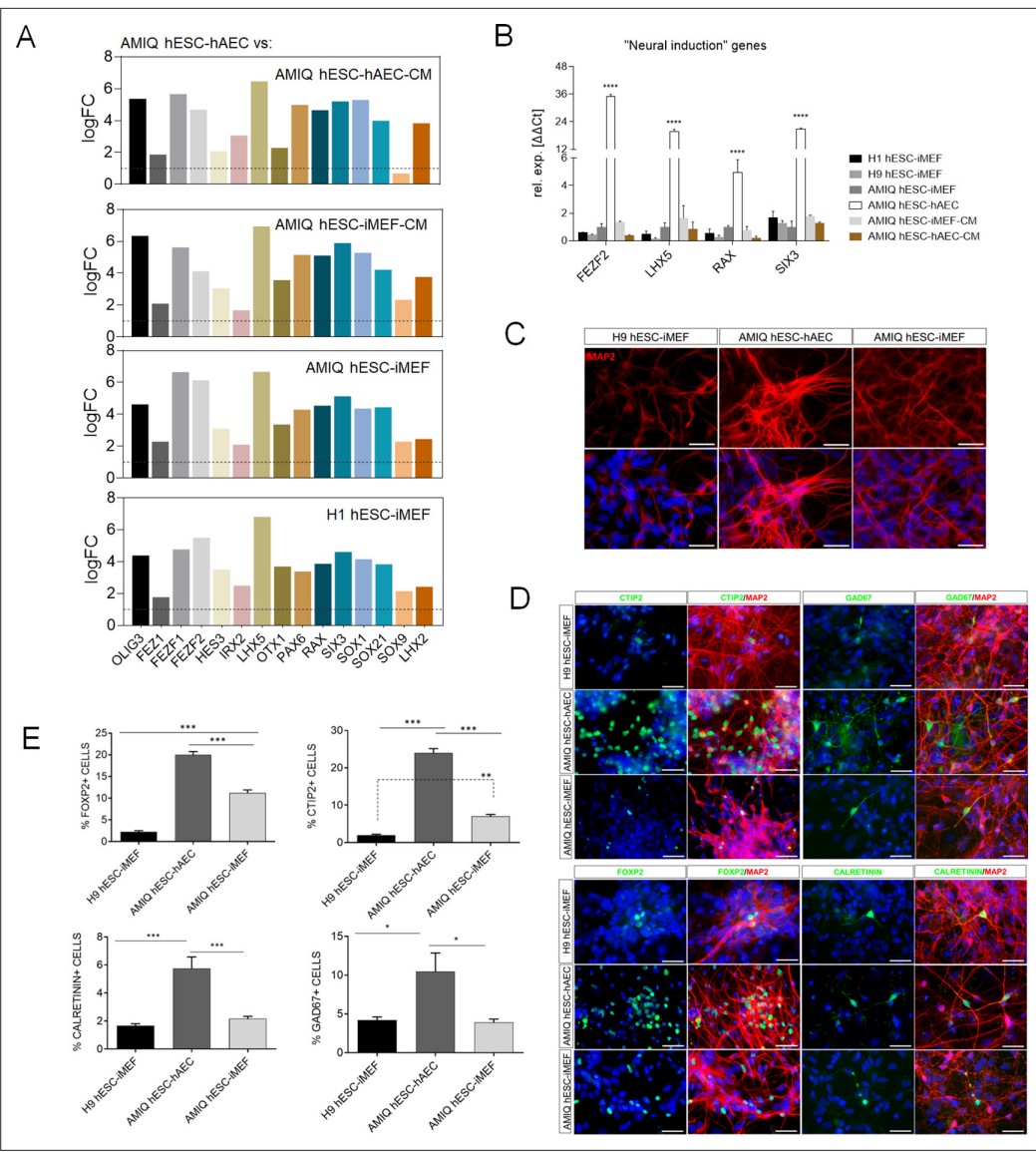

**Figure 5.** Upregulation of neural genes associated with forebrain development in human embryonic stem cell (hESC)-human amniotic epithelial cell (hAEC). (**A**) Expression of genes involved in forebrain development as signature molecular in Amicqui-1 hESC maintained on hAEC feeder layer (AMIQ hESC-hAEC). Log2 fold change determined using Limma-method from neural genes upregulated in AMIQ hESC-hAEC. FC>1.0 and p<0.05. (**B**) Validation by quantitative PCR (qPCR) array of selected transcripts upregulated in AMIQ hESC-hAEC. Data are mean ± SE and n=three biological samples per group. ****p<0.0001 using H1 hESC-inactivated mouse embryonic fibroblast (iMEF) as control. Lines and conditions used for (**A–B**) analysis: AMIQ hESC-hAEC, Amiqui hESC on alternative hAEC feeder layer; H1 hESC-iMEF, H1 hESC on conventional iMEF feeder layer; H9M hESC-iMEF, H9 hESC on conventional iMEF feeder layer; AMIQ hESC-iMEF, Amiqui hESC on conventional iMEF feeder layer; AMIQ hESC-iMEF-CM, feeder-free Amicqui-1 hESC with iMEF-conditioned media; AMIQ hESC-hAEC-CM, feeder-free Amicqui-1 hESC with hAEC-conditioned media. (**C**) Micrographs of the immunodetection of the mature neuron marker MAP2 in cells derived from H9 hESC-iMEF, AMIQ hESC-hAEC, and AMIQ hESC-iMEF on day 25 (D25). Scale bar, 50 μm. (**D**) Micrographs of the immunodetection of FOXP2+, CTIP2+, CALRETININ+, and GAD67+ neurons derived from H9 hESC-iMEF, AMIQ hESC-hAEC, and AMIQ hESC-iMEF. Scale bar, 50 μm. (**E**) Quantitative analysis of FOXP2+ or CTIP+ cells on day 45 (D45) and CALRETININ+ or GAD67+ on day 55 (D55). Data are mean ± SE. ** p<0.01 and *** p<0.001 with H9 hESC-iMEF as control. n=three biological samples per group and two repetitions per sample.

The online version of this article includes the following figure supplement(s) for figure 5:

*Figure 5 continued on next page*

*Figure 5 continued*

**Figure supplement 1.** Differential gene expression (DEG) analysis of genes involved in forebrain development as signature molecular AMIQ in human embryonic stem cell (hESC)-human amniotic epithelial cell (hAEC) compared with conventional, naïve, expanded, and XAV939 conditions.

**Figure supplement 2.** Interaction network and biological process (Gene ontology [GO]) of coding proteins of neural genes upregulated in AMIQ human embryonic stem cell (hESC)-human amniotic epithelial cell (hAEC).

**Figure supplement 3.** Micrographs of immunodetection of neural rosettes PAX6+/NESTIN+ derived from H9 human embryonic stem cell (hESC)-inactivated mouse embryonic fibroblast (iMEF), AMIQ hESC-human amniotic epithelial cell (hAEC), and AMIQ hESC-iMEF on day 16 (D16) of start the neural induction protocol.

**Figure supplement 4.** (**A**) Enrichment of transcription factors whose target genes (experimentally validated targets in the ChEA database) are upregulated in AMIQ hESC-hAEC compared to AMIQ hESC-iMEF.

**Figure supplement 5.** miR-200c relative expression in human embryonic stem cell (hESC) lines (Amicqui-1, H1, and H9) cultured on different feeder layers (inactivated mouse embryonic fibroblast [iMEF] or human amniotic epithelial cell [hAEC]) or feeder-free conditions (iMEF-CM or hAEC-CM).

---

cancer stem cell lines (*Iliopoulos et al., 2010*; *Peng et al., 2013*; *Simpson et al., 2021*). We found a high concentration of miR-200c-3p in hAEC and hESC-hAEC conditions compared with conventional hESC-iMEF (*Figure 5—figure supplement 5*). This data suggests that miR-200c-3p could influence the hESC-hAEC pluripotent state through SUZ12 inhibition to induce an NRG derepression.

The development (forebrain/regionalization) and type I hemidesmosome biological process were highly expressed in hESC-hAEC compared to hESC-hAEC-CM (*Figure 3—figure supplement 4D*). We postulated that NRG expression and neural induction enhancement could be a consequence of the embryonic cell-amniotic epithelium interaction instead of the molecule(s) secreted by the amnion. Proteome analyses demonstrated SRC, ERK2, STAT3, and AKT pathway activity in the AMIQ hESC-hAEC condition (*Figure 2*). Indeed, SRC is a master pathway that can activate ERK1/2, STAT3, and AKT (*Bjorge et al., 2011*).

Next, we inquired about the possible role of these pathways on the neural potential of AMIQ-hAEC co-culture. To this end, 48 hr before the neural induction protocol, we did treatment with PD0325901 (1 µM) for ERK2 (PD), LY294002 (5 µM and 10 µM) for PI3K/AKT (LY), Stattic (1 µM, 2.5 µM, and 5 µM) for STAT3, and A419259 (1 µM and 3 µM) for SRC inhibition (A41). The pluripotency markers and 'intermediate' and neural gene expression were checked before following the neural induction protocol. At the end of the protocol, the yield of NESTIN+/PAX6+ progenitors and MAP2+ neurons was analyzed.

We did not observe any differences in the OCT4+/NANOG+ colonies with the LY inhibitor in hESC-iMEF and hESC-hAEC conditions compared to the control (medium with FGF2). The A41 treatment diminished the OCT4+/NANOG+ cells in both situations, although the decrease was more significant in the hESC-hAEC condition, whereas PD treatment-induced pluripotency marker loss in the hESC-hAEC condition. Stattic had no effect in OCT4+/NANOG+ colonies in either situation (*Figure 6*).

OTX1, SALL2, SOX3, and POU3F1 expression in AMIQ-hESC-iMEF was not disturbed with the inhibitors. On the other hand, inhibition of SRC and ERK1 resulted in decreased OTX1 and SOX3, whereas PD treatment affected SALL2 and POU3F1 expression, and Stattic treatment only downregulated SALL2 expression in hESC-hAEC conditions (*Figure 6—figure supplement 1*).

Our results suggest a pluripotent network disruption promoted by SRC and ERK pathway inhibition in AMIQ-hAEC. Thus, we evaluated their differentiation potential under these challenging conditions. To this end, pre-treatments (48 hr) with the inhibitors were carried out (*Figure 7A*). None of the treatments (LY, A41, PD, and Stattic) induced changes in the expression of NGR signature on hESC-iMEF. In contrast, FEZF2, LHX5, RAX, and SIX3 expression with the SRC and ERK2 inhibitors was downregulated in hESC-hAEC, while Stattic treatment affected RAX and SIX3 expression (*Figure 7A*).

We also analyzed neural induction in both conditions (hESC on iMEF or hAEC) after 48 hr pre-treatments with inhibitors (*Figure 7B and C*). In AMIQ hESC-iMEF, the efficiency to generate progenitors (PAX6+/NESTIN+) and mature neurons (MAP2+) was only disrupted by STAT3 inhibition. Interestingly, there was a decrease in the formation of both progenitors and mature neurons in the hESC-hAEC condition with pre-treatments for A41, PD, and Stattic. These data suggest that the alteration in both ERK and SRC kinase pathways reduces neural bias potential, specifically in the hESC-hAEC condition.

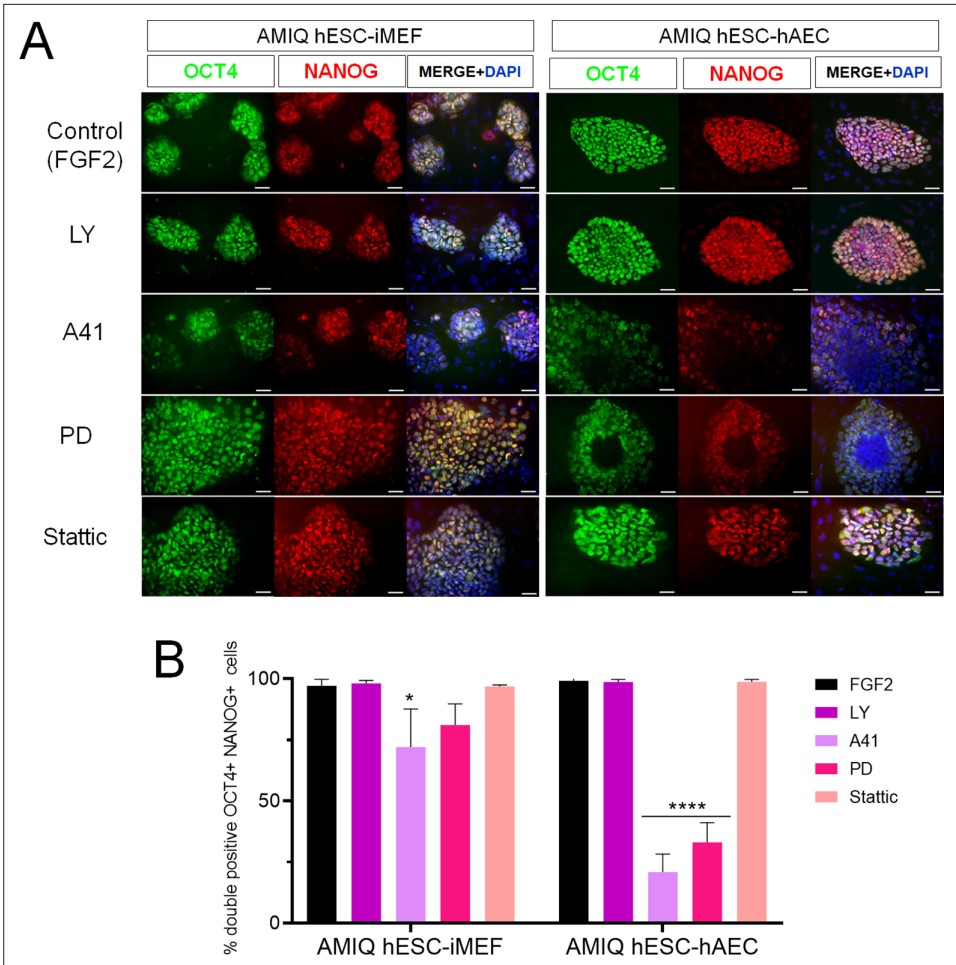

**Figure 6.** Inhibition of SRC and ERK2 disrupted expression pluripotency-related factors in human embryonic stem cell (hESC)-human amniotic epithelial cell (hAEC). (**A**) Immunofluorescence for the detection of NANOG+/OCT4+ colonies in hESC maintained on hAEC (AMIQ hESC-hAEC) and on inactivated mouse embryonic fibroblast (iMEF) (AMIQ hESC-iMEF) after treatment for 48 hr with inhibitors for specific signaling pathways. Control, hESC medium supplemented with dimethyl sulfoxide (DMSO) (1 μ/mL) and FGF2 (10 ng/mL); LY, hESC medium supplemented with FGF2 (10 ng/mL) and PI3K inhibitor LY294002 (5 μM); A41, hESC medium supplemented with FGF2 (10 ng/mL) and SRC inhibitor A419259 (1 μM); PD, hESC medium supplemented with FGF2 (10 ng/mL) and ERK2 inhibitor PD0325901 (1 μM); Stattic, hESC medium supplemented with FGF2 (10 ng/mL) and STAT3 inhibitor Stattic (2.5 μM). (**B**) Quantification of double positive OCT4+NANOG+ cells for each treatment in hESC on iMEF or hAEC. Data are mean ± SD. n=two biological samples per group and two repetitions per sample (20 colonies for each condition were analyzed). * p=0.0125 using AMIQ hESC-iMEF without any inhibitor as control and **** p<0.0001 using AMIQ hESC-hAEC without any inhibitor as control.

The online version of this article includes the following figure supplement(s) for figure 6:

**Figure supplement 1.** Analysis of formative pluripotency-related genes by quantitative PCR (qPCR) array in Amicqui-1 maintained on inactivated mouse embryonic fibroblast (iMEF) feeder layer (AMIQ human embryonic stem cell [hESC]-iMEF) or human amniotic epithelial cell (hAEC) feeder layer (AMIQ hESC-hAEC), under conditions of inhibition of specific signaling pathways for 48 hr.

During the neural induction protocol in the hESC-hAEC condition, the iSRC or iERK treated-cells presented a squamous epithelial morphology instead of neural rosettes and neural processes. Furthermore, the DEG analysis showed a high expression of epithelialization (TRIM29, CST6, and ITGB6) and identity of the amniotic ectoderm (VTCN1) genes in the hESC-hAEC condition (*Figure 4B*). Thus, we determined whether iSRC or iERK upregulated their expression. However, they showed a lower expression in the hESC-hAEC group (*Figure 7—figure supplement 1*).

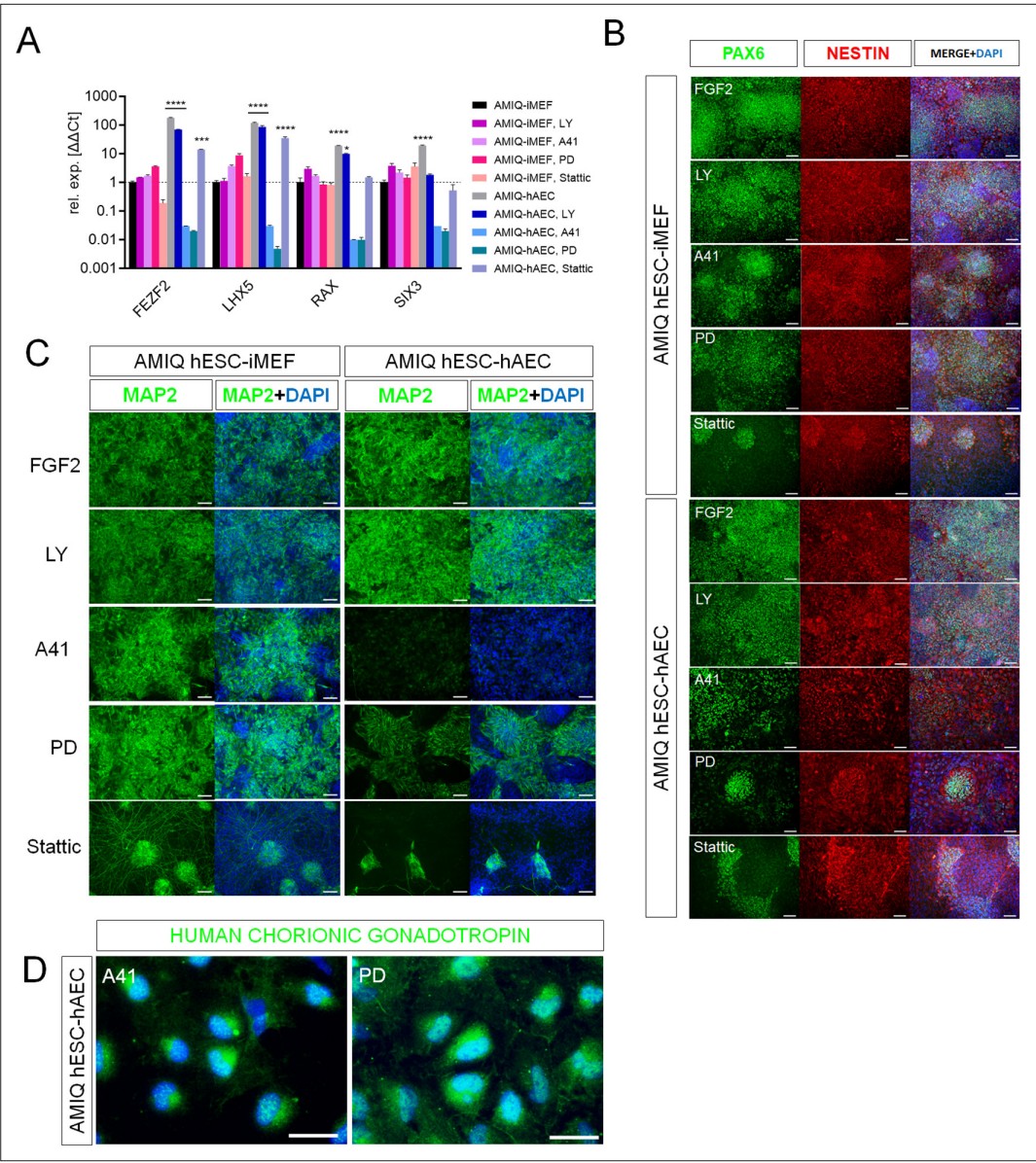

**Figure 7.** SRC and ERK2 inhibition alter biased neural potential of human embryonic stem cell (hESC)-human amniotic epithelial cell (hAEC). (**A**) Expression of selected neurodevelopmental regulatory gene (NRG) by quantitative PCR (qPCR) array in AMIQ hESC on inactivated mouse embryonic fibroblast (iMEF) (AMIQ iMEF) or hAEC (AMIQ hAEC) after 48 hr treatment with inhibitors of specific signaling pathways. Data are mean ± SD from technical duplicates. * p<0.05, ***p<0.001, and **** p<0.0001 with AMIQ hESC-iMEF as control. (**B**) Micrographs of immunodetection of neural rosettes PAX6+/NESTIN+ derived from AMIQ hESC-iMEF and AMIQ hESC-hAEC on day 20 of start the neural induction protocol and with a 48 hr pre-treatment with signaling pathway inhibitors. Scale bar, 25 µm. (**C**) Micrographs of immunodetection of mature neurons MAP2+ derived from AMIQ hESC-iMEF and AMIQ hESC-hAEC on day 30 of start the neural induction protocol and with a 48 hr pre-treatment with signaling pathway inhibitors. Scale bar, 25 µm. (**D**) Micrographs of immunodetection of human chorionic gonadotropin+ cells derived from AMIQ hESC-hAEC on day 30 of start the neural induction protocol and with a 48 hr pre-treatment with SRC and ERK2 signaling pathway inhibitors. Scale bar, 25 µm.

The online version of this article includes the following figure supplement(s) for figure 7:

**Figure supplement 1.** Analysis of 'intermediate genes' by quantitative PCR (qPCR) array in Amicqui-1 maintained on inactivated mouse embryonic fibroblast (iMEF) feeder layer (AMIQ human embryonic stem cell [hESC]-iMEF) or human amniotic epithelial cell (hAEC) feeder layer (AMIQ hESC-hAEC), under conditions of inhibition of specific signaling pathways for 48 hr.

*Figure 7 continued on next page*

*Figure 7 continued*

**Figure supplement 2.** (**A**) LogFC from differential gene expression (DEG) analysis (FC<1.0, p-value<0.05) for NR2F2 and VTCN1 genes in AMIQ human embryonic stem cell (hESC)-human amniotic epithelial cell (hAEC) compared with conventional primed, feeder-free, naïve, expanded, and XAV939 conditions as well as versus H9-derived amnion and hAEC isolated from membranes at term.

We also detected that, like VCTN1, another upregulated 'intermediate gene' was N2RF2 in DEG analyses of AMIQ hESC-hAEC (*Figure 7—figure supplement 2A*). Implantation occurs on the side of the embryo from which the polar trophectoderm arises, whose marker it is NR2F2 (*Meistermann et al., 2021*). After treatment with SRC or ERK signaling pathway inhibitors, NR2F2 expression was not significantly elevated, but 25 days after starting the neural induction protocol, its expression was even higher than in the initial condition of AMIQ hESC-hAEC (*Figure 7—figure supplement 2B*). We used immunofluorescence to detect amnion (KRT7) and trophectoderm (human chorionic gonadotropin, hGC). The cells were not positive for KRT7 (data not shown). Therefore, together with the data on the low expression of VCTN1, we would rule out that the inhibition of SRC or ERK induced the hESC on hAEC to differentiate toward the amniotic epithelium. However, we identified the presence of hGC, suggesting that they could have been determined into the trophectoderm in these populations.

## Discussion

DEG and GO analysis from transcriptome data showed that the hESC-hAEC molecular state was different from primed, naïve, expanded, XAV939, and 3iL hESC conditions, with several upregulated pathways (*Figure 3—figure supplement 2*) and biological processes involved in morphogenesis and development (*Figure 3—figure supplement 4*), including a specific gene set coding for transcription factors (NRG) involved in anterior neuroectoderm development (*Figure 5—figure supplement 2*). Notably, the hESC-hAEC condition had little to no downstream regulation of biological processes compared to the feeder-free conditions (hESC-iMEF-CM, hESC-hAEC-CM, and hESC-TesR1) and expanded conditions (*Figure 3—figure supplement 4B, C, D, F*). The most constant downregulated processes (versus conventional, 2iL, and XAV) were 'chromatin organization' and 'activation of HOX genes' (*Figure 3—figure supplement 4A and E, G*, *Figure 3—figure supplement 6*). The earliest expression of HOX gene occurred in the most posterior segment of the primitive streak, a mechanism conserved in murines, fish, amphibians, and birds (*Deschamps and Duboule, 2017*). The primary detection of OTX2 (*Figure 1G*), following forebrain development and other anterior structure genes (*Figure 5—figure supplement 2*), the downregulation of genes such as HOX (*Figure 3—figure supplement 6*) as well as genes involved in primitive streak establishment such as WNT3 (*Figure 3—figure supplement 3*), support the notion that the hESC-hAEC condition could represent transitional or regional pluripotency corresponding to the anteriorized epiblast and be more suitable for neural induction and the differentiation of anterior structures. hESC-hAEC showed a better capacity to produce TFAP2C+/STELLA+ and BLIMP1+/STELLA+ cells compared with hESC-iMEF (*Figure 4—figure supplement 1*). The origin of hPGC is not all clear. Its specification begins on day 12 post-fertilization, from an intermediate pluripotent subpopulation between naïve and primed, called 'germinal pluripotency' (*Chen et al., 2019*). On the other hand, it could also arise from the dorsal amnion at day 11 post-fertilization in cynomolgus monkeys (*Sasaki et al., 2016*). Indeed, the protocol to produce hPGC involves, as a first step, obtaining 'incipient MeLC' before expressing germline markers (*Sasaki et al., 2015*).

Furthermore, signaling from the amnion is required for mesoderm formation in non-human primates (*Yang, 2021*). Therefore, in the hESC-hAEC condition, the cells would acquire a greater competence to derive hPGC through non-embryonic mesoderm (they do not express some essential genes for establishing the primitive streak ) or an intermediate substate that generates both amnion and germline. However, this assessment needs further exploration.

Furthermore, the neural induction assay demonstrated that differentiation into cortical neurons (deep layers) was superior in AMIQ hESC-hAEC as compared with conventional hESC-iMEF, suggesting its preference to differentiate toward the anterior neuroectoderm (forebrain) (*Figure 5D and E*).

Which are the signals provided by the amnion (intracellular signaling cascades activated either by amnion-secreted factors that bind to hESC-membrane receptors or embryonic cell-amniotic cell

interactions) to trigger the slant toward the anterior ectoderm's fate in hESC? Although we did not identify a significant molecule by which hAEC induce their effect to maintain the unique molecular identity of AMIQ hESC-hAEC (in convergence with exogenous FGF2), the proteome arrays showed a contrasted activation of pluripotency-related pathways, which have also been identified in the early human and pig embryo, such as PI3K/AKT/mTOR signaling (*Wamaitha et al., 2020*; *Yilmaz et al., 2018*; *Zhou, 2009*) and JAK/STAT and MAPK paths (*Ramos-Ibeas et al., 2019*), respectively, as well as SRC family kinase signaling (*Figure 2*).

Differential response to the neural differentiation was evident with inhibitor treatments in hESC-hAEC compared with hESC-iMEF. First, PI3K/AKT inhibition did not affect either condition, while STAT3 inhibitor altered both conditions for neural induction efficiency. Lastly, SRC and MAPK blockade disrupted the biased neural potential of AMIQ hESC-hAEC, showing decreased NRG and fewer neural progenitors and mature neurons, unlike its AMIQ hESC-iMEF counterpart (*Figure 7A, B and C*).

Since the SRC pathway can trigger PI3K/AKT, JAK/STAT, and MAPK downstream signaling, it can be considered a master pathway. PI3K/AKT induces adhesion, migration, growth, and survival during hESC pluripotency maintenance. In contrast, MAPK and JAK/STAT could have an essential role in differentiation, implantation, and embryogenesis rather than in maintaining pluripotency. For example, the IL6/JAK/STAT3 pathway is involved in the epiblast-trophectoderm interaction during implantation. Our data suggest an interaction between amnion and embryonic cells involves the SRC and ERK2 pathway. Therefore, SRC can be triggered by EGFR with its ligand EGF or focal adhesions. The amnion secretes EGF (*Koizumi et al., 2000*), and the proteome showed a higher activity of EGFR in hESC-hAEC versus hESC-iMEF (*Figure 2A*).

Moreover, pathway enrichment analysis demonstrated an overrepresented focal adhesion pathway in hESC-hAEC (*Figure 3—figure supplement 2*), which may activate the SRC pathway. We did not find NRG upregulation in the hESC-hAEC-CM condition; it is more likely that the biased effect toward neural differentiation is due to the co-culture of both cell types. In AMIQ hESC-hAEC, the hemides-mosome and focal adhesion mechanotransduction signals were highly enriched, involved in adhesion and polarized migration and differentiation (*Jansen et al., 2017*; *Wang et al., 2020*). Therefore, this phenomenon would play an essential role in embryogenesis and morphogenesis. For example, it has been suggested that the polar trophectoderm exerts mechanical forces to mold the disc-shaped epiblast characteristic of primate embryos (*Weberling and Zernicka-Goetz, 2021*). Hence, we propose that the EGF pathway, a master pathway that activates PI3K/AKT, JAK/STAT, and MAPK pathways, and focal adhesion signals induce SRC-family activation.

These overrepresented paths could hinder the use of NRG transcriptional repression complexes in AMIQ hESC-hAEC, predisposing hESC to a pluripotent state directed toward lineages of anterior ectoderm-epiblast. For example, Akt can phosphorylate EZH2, displacing it from the PCR2 complex and switching from a repressor to a transcriptional activator on the previously silenced target genes (*Cha et al., 2005*; *Tan et al., 2014*).

On the other hand, interaction with amniotic epithelium upregulated a set of NRGs such as LHX5 and SIX3 in hESC (*Figure 5A*, *Figure 5B*, *Figure 5—figure supplement 1*), which are epiblast gene signatures in the pre-gastrula (*Tyser, 2021*). These NRGs induce cortical phenotypes, as we corroborated in our differentiation protocol toward neural fate (*Figure 5D and E*). However, temporary inhibition (pre-treatment 48 hr before neural induction) of SRC, ERK2, and STAT3 disrupted these neural biased (*Figure 7A–C*), and alternatively, the derived cells increased the expression of N2RF2 (*Figure 7—figure supplement 2*), so depending on the signals received, hESC also could differentiate into the polar trophectoderm.

To explain the NRG upregulation in the pluripotent state of hESC-hAEC, we hypothesized two mechanisms: (1) a PCR2 depression in target genes via phosphorylation of EZH2. EZH2 might promote their transcriptional activation together with cofactors promoted by IGF2/PI3K/Akt/mTOR and JAK/STAT3 pathways. The selective NRG upregulation might be mediated through antisense non-coding RNAs that function as guide RNAs to depress or even activate their complementary protein-coding sense partner. (2) hESC-hAEC have different pluripotent sub-states, of which one of them presents a bias toward the ectoderm, characterized by specific miRNAs enrichment. Moreover, subunits of PCR2, such as SUZ12, are inhibited by the miR-200 family in cancer stem cells. Interestingly, miR-200c-3p was highly expressed in hAEC and hESC-hAEC under the feeder layer and CM (*Figure 5—figure supplement 5*). Thus, miR-200c-3p is a candidate to regulate SUZ12 in NRG expressed in hESC-hAEC.

Additionally, miR-200c-3p could inhibit molecules involved in the epithelial-mesenchymal transition to avoid the differentiation into mesoderm lineages.

The contrast between hESC-hAEC and conventional hESC-iMEF in differentiating toward anterior neural destinations could be due to epiblast regionalization in the most anterior part. The imperative model of neural induction during embryogenesis in vertebrates establishes that neural cells have an anterior identity by 'default' and differentiate into a posterior fate in response to extrinsic signals. However, this notion has recently been challenged, proposing a period between the pluripotency stages and neural induction where the embryonic cells acquire a regional identity (anterior versus posterior) (*Metzis et al., 2018*). It has recently been demonstrated that the mouse epiblast presents regional and temporal states of pluripotency to generate the epiblast-ectoderm or the posterior epiblast at E6.6 (*Peng et al., 2019*). The primed pluripotency would be in the posterior region of the epiblast to form the primitive streak. In contrast, the formative pluripotency is maintained in the anterior epiblast by inhibiting pathways such as WNT. As has been proposed, this pluripotency substate would become restricted in the anterior epiblast to form the neuroectoderm (*Smith, 2017*), and we suggested that in humans the amnion might play an organizing role to maintain formative pluripotency or skew the pluripotent potential toward the neuroectoderm (*Figure 8*).

The anterior visceral endoderm (AVE) promotes the anterior regionalization and restricts the formation of the primitive streak to the posterior region of the epiblast in the mouse (*Hoshino et al., 2015*; *Yoshida et al., 2016*). On the other hand, the extraembryonic ectoderm promotes the induction and specific positioning of the AVE (*Richardson et al., 2006*; *Rodriguez et al., 2005*). Thus, the amniotic epithelium could be the equivalent to the extraembryonic ectoderm in primates and has a direct or indirect function in regulating the anterior regionalization of the epiblast (*Rossant and Tam, 2022*).

Indeed, hPSC can self-organize to develop an anteriorized embryonic-like sac where two types of cells co-exist: a population corresponding to the anterior epiblast-like cells and amniotic ectoderm-like cells. This suggests that in peri-implantation-stage human embryos, there is a lapse where the anterior epiblast co-exists with the nascent amniotic epithelium (*Zhang et al., 2014*).

Because our differential expression data were obtained from bulk RNA-seq, it was impossible to identify whether AMIQ-hESC-hAEC was composed of two coexisting subpopulations, a primed-formative pluripotent with potential skewed toward the anterior neuroectoderm and the other with potential to transit toward the amniotic squamous epithelium. Three-dimensional models of human embryogenesis can elucidate whether 'intermediate genes' are related to the amnion's specification from the epiblast and resolve the supposed importance of the amnion's interaction with the embryonic cells. We theorize that this interaction is essential to understand the molecular mechanisms and signaling pathways that could promote pluripotency regionalization with a predisposition to develop anteriorized structures such as the brain (*Figure 8*). It has been demonstrated that the cerebrospinal liquid arises from the amniotic fluid, which is arrested in the amniotic cavity when the neural tube is enclosed, providing evidence of the intimate relationship between the amnion and the onset of neuroectoderm induction in the anterior epiblast (*Chau, 2015*).

In the mouse embryo, the transition from naïve to primed epiblast is characterized by Wnt inhibition and OTX2 induction, so the epiblast forms a rosette with a lumen (*Neagu et al., 2020*). In humans, lumenogenesis forms the amniotic cavity during implantation. In this stage, the amnion derives from the epiblast. Thus, we propose that an epiblast subpopulation specified into the epithelium-amnion may be characteristic of gestation in humans and other primates. Moreover, hESC-hAEC expresses a set of genes described in an intermediate population between naïve and primed hPSC (*Figure 4B and C*; *Messmer et al., 2019*). VTCN1, DSC3, CERCAM, and ODAM are localized in epithelial phenotype cells; besides, the GO analysis of the AMIQ hESC-hAEC transcriptome indicated the upregulation of biological processes related to cell adhesion, epithelial development, and organization of the ECM, which agrees with the theory that the formative cells undergo epithelialization (*Figure 4—figure supplement 4*; *Smith, 2017*). Intriguingly, the 'intermediate gene' expression involved in epithelialization was higher in hAEC alone and in the hPSC-derived amnion than in the hESC-hAEC condition. We propose that 'intermediate genes' are an expression signature to monitor the transition from a columnar epiblast to the squamous amniotic epithelium. Thus, we inquired whether during the transition between naïve and primed human stages, a subpopulation of epiblast or embryonic cells has the signature of these genes and could participate in the amniotic epithelium specification.

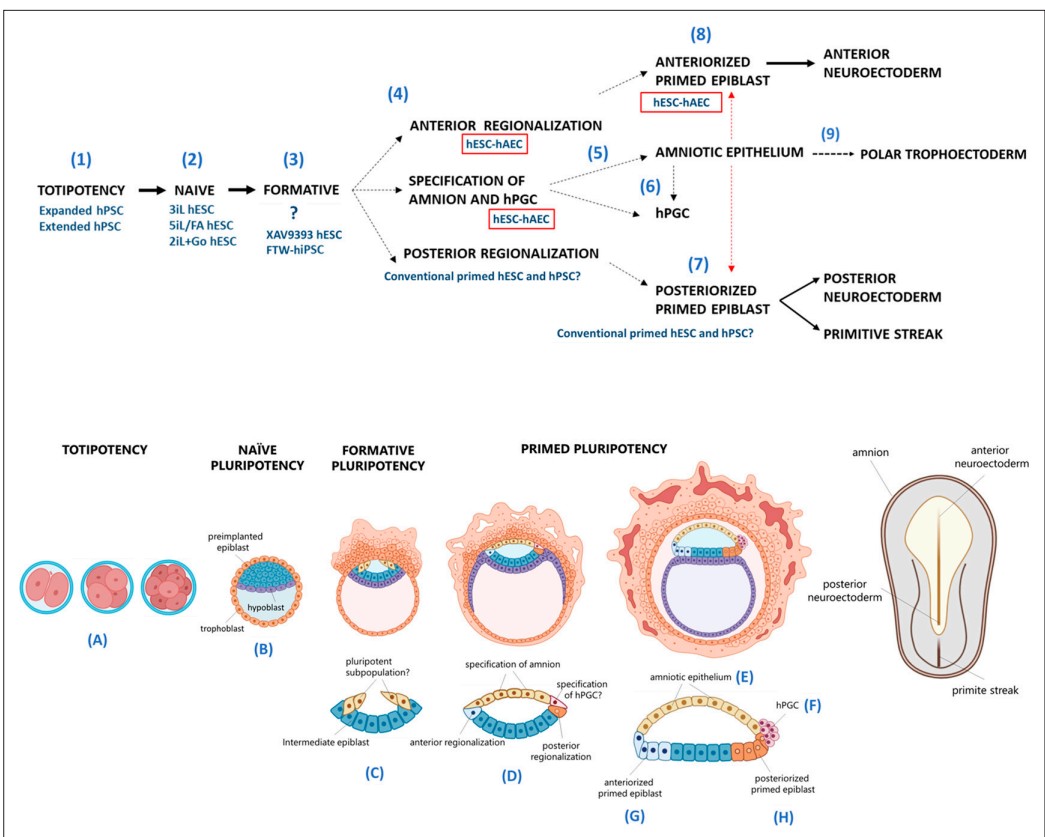

**Figure 8.** Proposed model for human embryogenesis with amnion as an organizer. Simplified scheme of embryonic stages in humans and its in vitro counterparts of human pluripotent stem cell lines (hPSC) represent or possess some characteristics of each state. Totipotency can be represented by the expanded and extended hPSC (1). Different conditions to maintaining naïve hPSC have been characterized, which resemble the pre-implantation epiblast (2). The formative pluripotency has recently been proposed as an intermediate period between the naïve and primed states. hPSC treated with the WNT inhibitor XAV939 and FTW-hiPSC could exhibit some characteristics of this phase, although there is still no established consensus of the in vitro conditions that represent this period (3). At peri-implantation, pluripotency could be spatially disaggregated for epiblast anterior-posterior regionalization and specification of amnion and human primordial germ cells (hPGC) (4). The amnion could arise from a subpopulation of pluripotent cells of the formative or primed epiblast (5). Primordial germ cells (PGCs) could be derived from another subpopulation (called 'germinal pluripotency' by *Chen et al., 2019*), although it has been suggested that they could also be specified from amnion (6). The anteriorized post-implanted epiblast will generate the anterior structures such as the neuroectoderm (7). The posteriorized post-implanted epiblast will give rise to the primitive streak and the most posterior region of the neuroectoderm (8). Recent reports suggest that the polar trophectoderm derives from pluripotent cells that transit through amnion-like cells (9). The condition of human embryonic stem cells growing on inactivated mouse embryonic fibroblasts (hESC-iMEF) could share more characteristics of the posterior epiblast given the interaction with a fibroblast cell type to favor the epithelium-mesenchyme transition. hESC-human amniotic epithelial cells (hAEC) have a more significant bias to differentiate into anterior neuroectoderm, but they also have the capacity to form germline cells and probably polar trophectoderm. Therefore, hESC-hAEC could be composed of subpopulations that form a continuum from the formative phase, as well as an intermediate population for specification of the amnion and germ line, to finally segregate to the anterior part of the epiblast. The amnion secretes signals or interacts directly with the primed epiblast until the onset of gastrulation promotes the neural induction or primitive streak formation (dotted arrow, probably route of differentiation; dotted red arrow, probably interactions between lineages).

Interestingly, amnion nascent-related VTCN1 and epithelialization genes were decreased in SRC and ERK2 inhibition conditions in hESC-hAEC (*Figure 7—figure supplement 1*). This data suggests that their inhibition avoided differentiation into neuroectoderm and non-neural/amniotic ectoderm. However, in AMIQ hESC-hAEC before and after being treated with SRC and ERK2 inhibitors, NR2F2 was upregulated, which is considered essential for the maturation of the polar trophectoderm

(*Figure 7—figure supplement 2*). The side of the embryo where the polar trophectoderm arises is the endometrium implantation site of the embryo (*Meistermann et al., 2021*). Thus, it is postulated that there is an epiblast-trophectoderm interaction to ensure correct implantation (*Weberling and Zernicka-Goetz, 2021*). Still, the role of the amnion is unknown, considering that during implantation, the amniotic cavity is formed, which would be a tripartite epiblast-amnion-trophectoderm relationship.

On the other hand, there is a fine line for identifying trophectoderm and amnion lineages in primate embryos. For example, a pre-print argues that in blastoids (human blastocyst-like structures derived from hPSC), the most external lineage cells do not form trophectoderm-like cells but are amnion-like cells (https://www.biorxiv.org/content/10.1101/2021.05.07.442980v3). Other reports suggest that trophectoderm-like cells can derive from hPSC but first must transit through an amnion-like state (https://doi.org/10.1101/2021.06.28.450118, *Chhabra and Warmflash, 2021*). Although the trophectoderm is specified when defining the pre-implantation blastocyst, the extraembryonic ectoderm derived from the amnion probably corresponds to the polar trophectoderm, necessary for embryo implantation.

A previous study suggested that primordial germ cells (PGCs) specified from the nascent amnion in the cynomolgus monkey (*Sasaki et al., 2016*), although a recent study indicates that progenitors of human PGCs present a gene expression signature of both amniotic and gastrulation lineages (*Chen et al., 2019*). In addition, the extraembryonic mesoderm is also closely related to the amnion (*Yang, 2021*). These results indicate that the amnion has a predominant role in developing extraembryonic lineages such as trophoectoderm, extraembryonic mesoderm, and germline specification. Our results in conjunction with these studies could be the basis to establish the amnion as a novel organizer for correct embryogenesis beyond implantation in humans and non-human primates.

## Materials and methods
### hAEC isolation for use as a feeder layer

Fetal membranes were obtained after receiving informed consent from women with a pregnancy delivered at term gestation (37–40 weeks) by cesarean section and without a history of infection. The membranes were transported to the laboratory to obtain hAEC under sterile conditions. The entire amnion was mechanically separated from the chorion, washed with saline solution to remove the erythrocytes, and incubated in trypsin/EDTA 0.05% (Gibco, 25300–054) for two periods of 45 min each at 37°C with gentle shaking. The enzymatic digestion was inactivated with inactive fetal bovine serum (FBS) qualified for stem cells (Gibco 10439–024) and centrifuged at 200 g for 10 min at room temperature. Isolated cells were resuspended in culture medium for hAEC (450 mL of high glucose-DMEM [Gibco, 12430–054], 50 mL of inactive FBS qualified for stem cells, 5 mL of non-essential amino acids 100× [Gibco, 11140–050], 5 mL of antibiotic-antimycotic 100× [Gibco, 15240–062], 5 mL of sodium pyruvate 100 nM [Gibco, 11360–070], 500 µL of 2-mercaptoethanol 1000× [Gibco, 21985–023], and 10 ng/mL EGF [Peprotech, AF-100–15]). hAEC were cultured at a density of $3 \times 10^4$/cm$^2$ at 37°C until reaching confluence (5–7 days). After, the cells were detached with trypsin/0.05% EDTA, centrifuged at 200 g for 5 min, resuspended in DMEM with 10% inactive FBS and 10% dimethyl sulfoxide (Sigma, D2650) at a concentration of $3 \times 10^6$ cells/mL/cryovial, and stored at liquid nitrogen. 48 hr before passaging hESC, a cryovial of hAEC was thawed in a 6-well plate with hAEC medium, using them as a feeder layer once they reached 80% confluence (36–48 hr).

### hESC culture under feeder layer condition

The in-house hESC lines derived Amicqui-1, commercial H1 (WiCell, WA01), and H9 (WiCell, WA09) were cultered as previously reported (*Ávila-González et al., 2015*; *Ávila-González et al., 2019*). Briefly, the hESC were maintained either on iMEF or alternative hAEC feeder layer with hESC medium (395 mL of DMEM/F-12 [11330–032], 20% KOSR, 5 mL of non-essential amino acids, and 910 µL of mercaptoethanol), supplemented with 10 ng/mL of FGF2 (Peprotech, 100-18B), and cultured in a hypoxic atmosphere (5% O2, 90% N2, and 5% CO2) at 37°C. The passages were carried out with Accutase (Gibco, A1110501). Mycoplasma contamination was not detected by the Mycoplasma SYBR Green PCR kit (Norgen Biotek) tested using a real-time thermocycler Rotor Gene Q (QIAGEN).

## hESC culture under feeder-free condition

To prepare Matrigel (Corning, 356234)-coated plates, the matrix was resuspended in DMEM/F-12 based on the dilution factor according to the lot; subsequently, 1 mL of diluted Matrigel was added to each well of a 6-well culture plate and incubated at least 1 hr at room temperature. To obtain CM from iMEF or hAEC, the cells were seeded in culture dishes (100 mm) with 10 mL of hESC medium without FGF2. The CM was collected every 24 hr for seven consecutive days and frozen at –80°C. For use, the thawed iMEF-CM or hAEC-CM was filtered through a membrane with a 0.2 μm pore diameter, and 10 ng/mL of FGF2 was added. The hESC colonies on iMEF or hAEC were mechanically detached and seeded on coated plates and cultured either with iMEF-CM or hAEC-CM supplemented with Y27632 (10 μM, final concentration) (Stem Cell Technologies, 72304).

## hESC culture under naïve conditions

Amicqui-1 were cultured on iMEF for at least 10 passages under two naïve conditions reported by *Duggal et al., 2015* and *Zimmerlin et al., 2016*. Duggal medium is composed of hESC medium (DMEM/F-12, 20% KOSR, non-essential amino acids and mercaptoethanol) supplemented with 10 ng/mL FGF2, 1000 U LIF (Peprotech, 300–05), 1 μM PD0325901 (Tocris, 4192), 3 μM CHIR99021 (Tocris, 4423), 10 μM forskolin (Stemgent, 04–0025), and 50 ng/mL acid ascorbic (Sigma-Aldrich, A4544). Zimmerlin medium is constituted by hESC medium supplemented with 1000 U LIF, 1 μM PD0325901, 3 μM CHIR99021, and 4 μM XAV939 (Tocris, 3748). The passages were carried out every 3–5 days using Accutase. Cells in passage 7 were used for detection of naïve pluripotency markers by qPCR.

## STR genotyping analysis

To verify the authenticity of hESC lines (Amicqui-1, H1, and H9), STR genotyping analysis was carried out. hESC were obtained from their feeder layer (iMEF or hAEC) with Accutase for 5 min at 37°C. The DNA was amplified by multiplex PCR, and the products were analyzed by capillary electrophoresis using Applied Biosystems 3500 Genetic Analyzer. For fragment analysis, we used the GeneMapper ID v5 software.

## Quantitative PCR

Total RNA was isolated using TRIzol (Life Technologies, 15596026) and treated with DNAse I (Zymo Research, E1010). To perform RNA extraction enriched with miRNAs, we included glycogen (Thermo Fisher, R0551) for the small RNAs precipitation.

RNA purity was determined by the 260/280 nm ratio (≥1.6) using a spectrophotometer (Nanodrop), and RNA integrity was determined by electrophoresis on 1% agarose gels. The cDNA was synthesized with Oligo dT using the Reverse Transcription System Kit (Promega, A3500). The qPCR amplification was generated with KAPA SYBR FAST qPCR Master Mix 2× (KM4100) in a Rotor-Gene Q 7000 thermocycler (Qiagen). To standardize the qPCR and validate the quantification method, we carried out a dynamic range for each gene using concentrations of cDNA (1000, 100, and 10 ng). Melting curves were performed for each reaction to ensure the amplification of a single product. The data were normalized with GAPDH, and relative expression was calculated by the 2-ΔΔCT method using conventional hESC on iMEF as calibrator samples. To quantify miRNA-200c-3p, the samples were normalized using Spike-In miR-39 from *Caenorhabditis elegans* as control, which was added to each sample before RNA extraction (*Lamadrid-Romero et al., 2018*).

## Immunocytochemistry

The cultured cells were fixed with 4% Paraformaldehyde and washed three times with PBS 1×. Permeabilization was done with Triton (Sigma, T8787) at 0.3% for 30 min, followed by blocking with 5% bovine serum albumin (BSA) (SIGMA, A8806) for 30 min. Cell were incubated overnight at 4°C with primary antibodies in blocking solution (*Supplementary file 5*). The next day, they were washed three times with 1× PBS, and secondary antibodies coupled to fluorophores were added (Alexa 488 [Thermo Fisher, A11029], Alexa 596 [Thermo Fisher, A11036]) in blocking solution to a concentration of 1:1000 and incubated for 1 hr at room temperature. Three washes were carried out with 1× PBS, and a solution with 4', 6-diamino-2-phenylindole was added in 1× PBS (5 μg/mL) (D1306, Thermo Fisher) to stain the nuclei. As negative controls, hESC samples were incubated only with the secondary

antibody. Stains were visualized under an epifluorescence microscope (Olympus IX-81), and photographs were taken with a CCD camera (Hamamatsu, ORCA-Flash).

## Electrophoresis and western blot

To extract protein, culture cells were resuspended in lysis buffer containing protease and phosphatase inhibitors. The samples were centrifuged at 13,000 g for 10 min at 4°C, and the supernatant was collected to quantify the protein by Bradford assay. Protein (15 µg) from each sample was resolved on 12.5% SDS-PAGE gels and transferred to nitrocellulose membranes using the Trans-Blot semi-dry transfer cell system (Bio-Rad). For western blot detection, the membranes were incubated with Intercept (PBS) Blocking Buffer (LI-COR Bioscience) for 1 hr at room temperature with gentle shaking and incubated overnight at 4°C with the primary antibodies against OCT4 and NANOG using GAPDH as internal control (List of antibodies in *Supplementary file 5*). Membranes were rinsed with 1× PBS with 0.1% Tween 20 to incubate with the LI-COR secondary antibodies (List of antibodies) for 1 hr at room temperature. Scanning of the membranes was carried out with an Odyssey CLx System (LI-COR Bioscience), and the fluorescence intensity was analyzed with Image Studio version 4.0 software (LI-COR Bioscience).

## Determination of pluripotency in hESC with Epi-Pluri-Score

gDNA was isolated using the High Pure FFPET DNA Isolation Kit according to the supplier's instructions (Roche, 11732668001). The gDNA samples were sent to Germany (Cygenia, https://www.cygenia.com/) to analyze the DNA methylation at three specific CpG sites (ANKRD46, C14orf115, and POU5F1). The difference in DNA methylation levels (β-values) of the CpGs in ANKRD46 and C14orf115 was determined and combined as Epi-Pluri-Score, while the CpG site in POU5F1 further discriminated pluripotent from non-pluripotent cells (*Lenz et al., 2015*). The results of the Epi-Pluri-Score were represented in a scheme where the samples were classified and grouped according a base of pluripotent or non-pluripotent cell data.

## Proteome cytokine and phospho-kinase arrays

Identification of released cytokines and phospho-kinases in hESC lines (Amicqui-1 on MEF or HAEC and H1 on MEF) and hAEC cultured with hESC medium was carried out using Proteome Profiler Human Cytokine Array (R&D Systems, ARY022B) and Proteome Profiler Human Phospho-Kinase Array Kit (R&D Systems, ARY003B) following the manufacturer's instructions. For the cytokines, the supernatant of near confluent cultures maintained for 24 hr was collected and filtered through a membrane of 0.2 µm pore diameter, while the cell lysates were used for phospho-kinase profile detection. The spot densities of the arrays were analyzed with ImageJ software, and the average density of spots was normalized using H1 on iMEF as control.

## RNA-seq and bioinformatic analysis

Total RNA was extracted and DNAse I treated using Quick-RNA Miniprep Plus Kit (Zymo Research, R1058) according to the manufacturer's protocol. Concentration of the total extracted RNA was determined with a Qubit 2.0 (Invitrogen) (Life Technologies), while the quality of RNA samples was determined through a BioAnalyzer (Agilent). Only samples with an RNA integrity number >8.0 were used. Libraries were constructed using 500 ng of RNA, using the Truseq Stranded mRNA library prep kit from Illumina according to the manufacturer's instruction. Transcriptome sequencing was conducted in the Ilumina HiSeq 2500 platform with a configuration of 300 cycles to generate single and paired end reads of 2 × 125 read length. Raws were uploaded to the BioJupies platform (*Torre et al., 2018*) (https://amp.pharm.mssm.edu/biojupies/) to obtain DEG analysis. We considered a gene as differentially expressed with the log fold change and p cutoff values of 1.5 and 0.05, respectively. Overrepresentation of KEGG pathways in hESC-hAEC versus the other conditions was determined through the Pathview web using default parameters (p-value<0.05) (*Luo et al., 2017*). G:Profiler (*Raudvere et al., 2019*) (https://biit.cs.ut.ee/gprofiler/gost) tools were utilized for gene set enrichment analysis (biological processes and Reactome) from the list of upregulated and downregulated genes in hESC-hAEC versus the other conditions using default parameters (p-value<0.05). Visualization of the enrichment analysis obtained in G:Profiler was carried out using Cytoscape (p<0.01), and predicted protein-protein associated network was constructed from the

top of upregulated genes in hESC-hAEC with the latest version of STRING database (https://string-db.org/) (*Szklarczyk et al., 2019*). The enrichment of the transcription factors whose targets are differentially expressed hESC-hAEC genes was obtained by Enrichr (FDR <0.05) (*Kuleshov et al., 2016*). PCA of bulk RNA-seq expression data from cell lines was performed using the R 'prcomp' function based on log2-transformed Z-score values. To compare cell line bulk RNA-seq with human embryo single cell RNA-seq data, log counts per million transformed counts were used, and log2-transformed Z-score was calculated. The principal components of all samples were calculated with R 'prcomp' function. To merge the different datasets, the common genes were selected, and all those with the value of zero in all samples were eliminated. The most variable genes across the samples were plotted.

## Validation of RNA-seq data by qPCR-based array analysis

An RT2 PCR Array (Qiagen, 330171) was customized for the detection of 12 selected genes that were the most upregulated in hESC-hAEC. Previously isolated RNA samples were used to synthesize cDNA using the RT2 First Strand Kit (Qiagen, 330401), which were mixed with RT2 SYBR Green ROX qPCR MasterMix to load them into RT2 Profiler PCR Arrays. The samples were run in a Rotor-Gene Q 7000 thermocycler (Qiagen) under the following cycling conditions: one cycle of 10 min at 95°C and then 40 cycles of 15 and 30 s at 95°C and 60°C, respectively. The threshold cycle (CT) for each sample was calculated using the real-time cycler software. The data were normalized with five house-keeping genes (GUSB, GAPDH, PGK1, TFRC, and ACTB), and the relative expression was calculated by 2-ΔΔCT method, using hESC-iMEF (Amicqui line) as a calibrator.

## Induction of human primordial germinal-like cells

Two hundred thousand cells per well were plated on Matrigel-coated 12-well plates in GK15 medium: Glasgow media (Sigma-Aldrich, G6148) supplemented with 15% KSR, 0.1 mM NEAA, 2 mM L-gluta-mine, 1 mM sodium pyruvate, 0.1 mM 2-mercaptoethanol, 3 µM CHIR99021, 50 ng/mL Activin A (R&D Systems, 338-AC), and 10 µM ROCK inhibitor (StemCell Technologies, 72302). After 24 hr, cells were detached using Accutase and plated 4000 cells/well of GK15 medium supplemented with 200 ng/mL BMP4 (R&D Systems, 314 BP), 20 ng/mL human LIF, 100 ng/mL SCF (R&D Systems, 255-SC), 50 ng/mL EGF, and 10 µM ROCK inhibitor. Cells were fixed and analyzed on D9 and D12 to identify germline markers by immunofluorescence. The quantification was carried out as previously reported (*García-Castro et al., 2015*).

## Neural induction and cortical differentiation protocol

At 70% confluency, H9 and Amicqui-1 on conventional iMEF and Amicqui-1 on hAEC were cultured with hESC medium supplemented with SB431542 (10 Mm, final concentration) (R&D, 1640) and dorsomorphin (2 µM, final concentration) (R&D, 3092). The medium was changed daily, and on day 4 of the induction protocol (D4), the hESC were cultured with N2 medium (480 mL of DMEM/F12, 5 mL of N-2 supplement 100× [Gibco, 17502–048], 5 mL of GlutaMAX [Gibco, 35050–061], 5 mL of amino acid non-essentials, 5 mL sodium pyruvate, 500 µg/mL of BSA, 5 mL of penicillin-streptomycin [Gibco, 15070–063], 910 µL of mercaptoethanol, SB431542 [10 µM], and dorsomorphin [2 µM]). At D12, the cells were detached with 0.875 mg/mL Dispase II (Gibco, 17105–041) and plated in dishes treated with poly-L-ornithine and laminin (Sigma-Aldrich) at a density of 50,000 cells/cm$^3$ in N2 medium supplemented with 20 ng/mL FGF2, 20 ng/mL EGF, and Y27632 (10 µM, final concentration). The next day (D13) the medium was exchanged for a mixture (1:1) of N2 medium with B27 medium (490 mL of Neurobasal, 10 mL of B27 supplement 50× [Gibco, 12,587–010], 5 mL of GlutaMax, and 5 mL of penicillin-streptomycin) supplemented with FGF2 (20 ng/mL), and EGF (20 ng/mL). The following 3 days, N2B27 medium was changed daily, adding growth factors. From D16, growth factor treatment was suspended, and the N2B27 medium was exchanged every third day. At D20, the cells were passaged onto Matrigel-coated plates exchanging N2B27 medium daily. The cells were passaged again on D25 and D35. From this stage, the culture cells were maintained with N2B27 exchanged every 4–5 days until subsequent fixation and detection of neural and cortical lineage markers by immunofluorescence (D40–D60).

## Inhibition of signaling pathways by small molecules

Amicqui-1 on iMEF or hAEC was treated separately with LY294002 (StemCell Technologies, 72152) (5 µM and 10 µM), PD0325901 (Tocris, 4192) (1 µM), A419259 (Tocris, 3914) (1 µM and 3 µM), or Stattic (Tocris, 2798) (2.5 µM and 5 µM). Each group was maintained with hESC medium supplemented with FGF2 (10 ng/mL). The treatments were done 48 hr before the neural induction protocol.

## Acknowledgements

We thank Jessica González and Susana Castro-Chavira for their excellent technical assistance; Ricardo Grande, Veronica Jimenez-Jacinto and Alejandro Sanchez-Flores for Bioinformatics support that was performed in the Unidad de Secuenciación Masiva y Bioinformatica at the Laboratorio Nacional de Apoyo Tecnologico a las Ciencias Genomicas CONACYT 260481, Instituto de Biotecnologia, Universidad Nacional Autónoma de México (UNAM); and Rosa Rebollar-Vega for the support to constructing the libraries and sequencing at the Red de Apoyo a la Investigación- Coordinación de la Investigación Científica-UNAM. Daniela Ávila-González is a beneficiary of a CONACYT postdoctoral fellowship. Funding our research was supported by grants from Instituto Nacional de Perinatologia de Mexico (21041, 21081 and 2019-1-40), CONACYT (130627, 252756, 300638, 271307 and A1-S-8450) and FODECIJAL 8084–2019.

## Additional information

### Funding

| Funder | Grant reference number | Author |
| --- | --- | --- |
| Instituto Nacional de Perinatología | 21041 | Néstor F Díaz |
| Instituto Nacional de Perinatología | 21081 | Néstor F Díaz |
| Instituto Nacional de Perinatología | 2019-1-40 | Néstor F Díaz |
| Consejo Nacional de Ciencia y Tecnología | 130627 | Néstor F Díaz |
| Consejo Nacional de Ciencia y Tecnología | 252756 | Néstor Emmanuel Díaz-Martínez |
| Consejo Nacional de Ciencia y Tecnología | A1-S-8450 | Néstor Emmanuel Díaz-Martínez Néstor F Díaz |
| Consejo Nacional de Ciencia y Tecnología | 300638 | Néstor Emmanuel Díaz-Martínez |
| Consejo Nacional de Ciencia y Tecnología | 271307 | Néstor Emmanuel Díaz-Martínez |
| FONDECIJAL | 8084-2019 | Néstor Emmanuel Díaz-Martínez |

The funders had no role in study design, data collection and interpretation, or the decision to submit the work for publication.

### Author contributions

Daniela Ávila-González, Conceptualization, Data curation, Formal analysis, Investigation, Writing – original draft, Writing – review and editing; Wendy Portillo, Resources, Supervision, Validation, Methodology, Writing – review and editing; Carla P Barragán-Álvarez, Resources, Formal analysis, Supervision, Validation, Investigation, Methodology, Writing – review and editing; Georgina Hernandez-Montes, Resources, Data curation, Formal analysis, Supervision, Funding acquisition, Validation, Investigation, Methodology, Writing – review and editing; Eliezer Flores-Garza, Data curation, Formal analysis, Validation, Investigation; Anayansi Molina-Hernández, Néstor Emmanuel

Díaz-Martínez, Conceptualization, Resources, Formal analysis, Supervision, Funding acquisition, Validation, Investigation, Methodology, Writing – original draft, Writing – review and editing; Néstor F Díaz, Conceptualization, Resources, Data curation, Formal analysis, Supervision, Funding acquisition, Validation, Investigation, Methodology, Writing – original draft, Writing – review and editing

### Author ORCIDs
Daniela Ávila-González  http://orcid.org/0000-0003-4817-6630
Néstor F Díaz  http://orcid.org/0000-0003-2436-9374

### Decision letter and Author response
Decision letter https://doi.org/10.7554/eLife.68035.sa1
Author response https://doi.org/10.7554/eLife.68035.sa2

## Additional files

### Supplementary files
• Supplementary file 1. List of genes with differential expression in AMIQ human embryonic stem cells (hESC)-human amniotic epithelial cells (hAEC) compared to other culture conditions.

• Supplementary file 2. KEGG pathways upregulated in AMIQ human embryonic stem cells (hESC)-human amniotic epithelial cells (hAEC).

• Supplementary file 3. 'Intermediate genes' upregulated in AMIQ human embryonic stem cells (hESC)-human amniotic epithelial cells (hAEC).

• Supplementary file 4. Transcription factors whose targets are overrepresented in the upregulated and downregulated genes identified in AMIQ human embryonic stem cells (hESC)-human amniotic epithelial cells (hAEC).

• Supplementary file 5. List of antibodies used in this study.

• Transparent reporting form

### Data availability
RNA-seq data from Amiqui-1 are available at the ArrayExpress database (https://www.ebi.ac.uk/array-express/experiments/E-MTAB-10347/) under accession number E-MTAB-10347. Source code for the following analyses is available as indicated: BioJupies, https://github.com/MaayanLab/biojupies, copy archived at swh:1:rev:0b8b125740f742689b0cc3513db09b9f48866805.

The following dataset was generated:

| Author(s) | Year | Dataset title | Dataset URL | Database and Identifier |
|---|---|---|---|---|
| Ávila-González D, Portillo W, Molina-Hernández A, Díaz-Martínez NE, Díaz NF | 2022 | The amniotic epithelium triggers an alternative state of human pluripotency | https://www.ebi.ac.uk/biostudies/arrayexpress/studies/E-MTAB-10347 | ArrayExpress database, E-MTAB-10347 |

The following previously published datasets were used:

| Author(s) | Year | Dataset title | Dataset URL | Database and Identifier |
|---|---|---|---|---|
| Shao Y, Taniguchi K, Gurdziel K, Townshend RF, Gumucio DL, Fu J | 2016 | Self-organized amniogenesis by human pluripotent stem cells in a biomimetic implantation-like niche | https://www.ebi.ac.uk/ena/browser/view/PRJNA352339 | European Nucleotide Archive, PRJNA352339 |

*Continued on next page*

*Continued*

| Author(s) | Year | Dataset title | Dataset URL | Database and Identifier |
|---|---|---|---|---|
| Chan YS, Göke J, Ng JH, Lu X, Gonzales KA, Tan CP, Tng WQ, Hong ZZ, Lim YS, Ng HH | 2014 | RNA-Seq of 3iL human embryonic stem cells with an induced expression profile that more closely resembles native pluripotent cells and uninduced controls | https://www.ebi. ac.uk/biostudies/ arrayexpress/studies/ E-MTAB-2031 | ArrayExpress, E-MTAB-2031 |
| Guo G, von Meyenn F, Rostovskaya M, Clarke J, Dietmann S, Baker D, Sahakyan A, Myers S, Bertone P, Reik W, Plath K, Smith A | 2017 | RNA-seq of chemically reset human naive pluripotent stem cells | https://www.ebi. ac.uk/biostudies/ arrayexpress/studies/ E-MTAB-5674 | ArrayExpress, E-MTAB-5674 |
| Collier A, Panula S, Lanner F, Rugg-Gunn P | 2017 | Profiling human naive and primed pluripotent states | https://www.ebi.ac. uk/ena/browser/view/ PRJNA360413 | European Nucleotide Archive, PRJNA360413 |
| Smith AG, Rostovskaya M, Stirparo GG | 2019 | Formative transition of human naïve pluripotent stem cells | https://www.ebi.ac. uk/ena/browser/view/ PRJNA507424 | European Nucleotide Archive, PRJNA507424 |
| Gao X, Nowak-Imialek M, Chen X | 2019 | Establishment of Porcine and Human Expanded Potential Stem Cells (bulk RNA-seq) | https://www.ebi. ac.uk/biostudies/ arrayexpress/studies/ E-MTAB-7253 | ArrayExpress, E-MTAB-7253 |
| Yu L, Wei Y, Sun H, Mahdi A, Sakurai M, Pinzon CA, Okamura D, Mutto AA, Li J, Gu Y, Ross P, Wu J | 2020 | Derivation of formative like pluripotent stem cells from mamalian embryos (RNA-Seq) | https://www.ncbi. nlm.nih.gov/geo/ query/acc.cgi?acc= GSE135989 | NCBI Gene Expression Omnibus, GSE135989 |

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
