## [Editor Report]

A number of different stages of pluripotency have been described in the literature. Ávila-González et al., show that human amniotic epithelial cells (hAECs) can be used to support a unique primed pluripotent state in which cells are biased towards an anterior neural identity. The authors explore the unique identity of these lineage-primed pluripotent cells suggesting they could be used for more efficient neural differentiation. The ability of hAECs to support a pre-patterned cell type that continues to express pluripotency markers also implies new roles for the aminion in patterning and supporting early human embryos.

---

## [Decision Letter]

**Decision letter after peer review:**

Thank you for submitting your article "The amniotic epithelium triggers an alternative state of human pluripotency" for consideration by *eLife*. Your article has been reviewed by 3 peer reviewers, and the evaluation has been overseen by a Reviewing Editor and Marianne Bronner as the Senior Editor. The reviewers have opted to remain anonymous.

Essential revisions:

1) Better evidence that these cells represent a new pluripotent state. All three reviewers would like to see a better indication of what defines this state, improved gene expression analysis, in vivo comparsons and analysis of gene expression. What is the profile of hAEC vs hESC in these co-cultures? Two of the reviewers felt that the existing RNAseq analysis takes up large sections of the main figures, but add little to the narrative.

2) Some evidence that molecule(s) produced by hAEC cells can induce this pluripotent state and/or the mechanism by which they induce it would be essential for publication in *eLife*.

3) While the authors have focused on enhanced anterior neural specification, the notion that this is biased or regionalized state of pluripotency, would also suggest that differentiation to other lineages might be restricted. They should compare the relative efficiencies with which hAEC cultured hESCs and standard hESCs differentiate into primitive streak derivatives or primordial germ cells.

*Reviewer #1 (Recommendations for the authors):*

González et al., chacterized the hAEC culture condition and found it was dependent on exogeneous FGF2 and then characterized specific signalling pathways there were unique to the hAEC supported pluripotency and cytokines produced by hAECs. In particular they observed an abundance of Src activation in hAEC supported ESCs. They follow this up with transcriptome analysis, where they compare hAEC supported cells to different published ESC culture systems. There are a lot of comparisons, but the bottom line is these cells appear anterior epiblast poised for neural differentiation. While the generation of neural progentintors from all lineages tested was similar, the hAEC cultured cells gave rise to enhanced differentiation towards neural lineages found in the forebrain. They conclude by suggesting hAEC-cultured ESCs exhibit upregulation of polycomb targets and propose that hAEC cells produce a factor that could repress polycomb activity. They suggest that this mechanism might by mediated by miR200b/c miRNA, highly expressed in hAEC and upregrulated in hESCs cultured with hAEC.

As suggested above, this paper suggests hAEC impart regionalation or developmental identity on hESC cultures, produces cells that resemble the anterior epiblast that is uniquely poised for anterior neural differentiation. The paper could be strengthened by providing a better assessment of the cell states formed by co-culture with hAECs, how do they cells compare to available primate RNAseq datasets and how homogeneously are they anterior epiblast? In a similar set of analysis they could address the nature of the hAECs used here and explore whether they represent regionalized amnion? In addition the paper suggests candidates for the molecules produced by hAECs that induce this phenotype, but they do not provide any functional evidence for their candidates.

This manuscript introduces a novel idea, that amnionic epithelial cells can instill positional epiblast identity onto human ESC cultures. However, there remain a number of issues that should be resolved prior to publication in *eLife*.

1. The gene expression analysis in Figure 4-6 is incomplete and yet takes up a large part of the main figures. If these cells generally represent regionalized, pre-neural epiblast, normally found adjacent to the amnion, then some analysis of the single cell sequencing from the human and primate embryo data should be undertaken (Xiang et al., 2020; Niu et al., 2019; Nakamura et al., 2017), this analysis should also include the spectrum of ESC cell states that have been published. The authors perform a number of comparisons, but none of these place the hAEC supported cells in context of this spectrum; they could for example compare gene expression changes that occur as development progresses to the gene expression changes that occur in the different ESC cell states as they progress or alternatively try some sort of global analysis of developmental stage with the different ESC data.

2. The model in 8b implies that the amnionic epithelium has regionalized inductive properties. What cell type is represented by the hAEC cells. Using similar analysis to that described above, they could address this issue. It also might help them to refine their list of potential "inductive," molecules.

3. There is a lot of discussion of potential candidates for induction of anterior epiblast fate by hAECs, but none of these are tested. To produce the level of advance required for *eLife*, some demonstration of a candidate produced by the hAEC cells that promotes anterior epiblast identity in ESCs, is important. Even better, if the authors could find a candidate that was also localized in the amnion based on the single cell transcriptome data and that was produced by their hAEC cells this would make the manuscript much stronger.

4. The demonstration that these cells are poised for anterior neural specification is very interesting. Are they less effective at generating primitive streak cell types?

5. There appears some confusion in the text about the term pluripotency. It is not really pluripotency that becomes regionalized, but epiblast. Or one could generate pluripotent cells with regional identity. The is also a bit of confusion between different stages of pluripotency. For example, the so called "Rosette," state and "Formative," state. The Rosette state requires only inhibition of Wnt signaling (Neagu et al., 2020), while Formative pluripotency is achieved based on relief of FGF/ERK inhibition as well (Kalkan et al., 2017) and trapped by Wnt and RAR inhibition(Kinoshita et al., 2021), and it is not clear how different "formative," is from some variations of standard, primed human ESCs (Davidson, Mason, and Pera 2015). However, anterior neural differentiation can be directly accessed in the presence of a block to ERK from naïve pluripotency, suggesting it can occur via something like a Rosette state or pre-formative epiblast (Hamilton and Brickman 2014). When and how does anterior epiblast become primed for differentiation away from primitive streak derivatives? This is an excellent question, but the data in the paper doesn't really address this. As it is difficult to resolve so called "formative," markers from anterior epiblast, perhaps it would be easier to avoid overuse of specific labels and stick with analogous embryonic structures. While these cells are FGF2 dependent, it would suggest they are more "primed," rather than formative but as this is all in a human context it is difficult to resolve this and as eluded to above primed pluripotency in some human examples maybe closer to what is called formative. Moreover, I also find it hard to resolve "formative," and anterior epiblast, based on the handle of markers assessed here. Perhaps they will be able to derive better evidence for this staged model of differentiation from the transcriptomic analysis requested in point 2, but I am not sure this will provide a clear answer. Experimentally they could ask how dependent these cells on ERK signaling? This would be interesting to resolve, as ERK activity is significantly lower in the anterior epiblast than on the posterior side in the region of the primitive streak. They could attempt to convert naïve cells to this state through a staged differentiation trajectory with specific inhibitors and demonstrate when these cells become uniquely anteriorized? While these are all interesting experiments, they could also just revise the discussion and Figure 8 and focus on epiblast-like, anterior epiblast-like.

Specific Comments

1. Otx2 upregulated in AEC feeder culture, but not with conditioned media (Figure 2B)? As Otx2 is one of the principle markers interpreted, perhaps the authors should give this some thought.

2. Increase size of labels on the figures, they are very difficult to read.

3. hAECs have some pluripotent gene expression and qualities. What is the extent that the data in Figure 3C, reflects the activity in hAECs or hESCs? Should these experiments not be using conditioned media?

References

Davidson, Kathryn C, Elizabeth A Mason, and Martin F Pera. 2015. 'The Pluripotent State in Mouse and Human.' Development 142 (18): 3090-99. https://doi.org/10.1242/dev.116061.

Hamilton, William B., and Joshua M. Brickman. 2014. 'Erk Signaling Suppresses Embryonic Stem Cell Self-Renewal to Specify Endoderm'. Cell Reports 9 (6): 2056-70. https://doi.org/10.1016/j.celrep.2014.11.032.

Kalkan, Tüzer, Nelly Olova, Mila Roode, Carla Mulas, Heather J. Lee, Isabelle Nett, Hendrik Marks, et al. 2017. 'Tracking the Embryonic Stem Cell Transition from Ground State Pluripotency'. Development 144 (7): 1221-34. https://doi.org/10.1242/dev.142711.

Kinoshita, Masaki, Michael Barber, William Mansfield, Yingzhi Cui, Daniel Spindlow, Giuliano Giuseppe Stirparo, Sabine Dietmann, Jennifer Nichols, and Austin Smith. 2021. 'Capture of Mouse and Human Stem Cells with Features of Formative Pluripotency'. Cell Stem Cell 28 (3): 453-471.e8. https://doi.org/10.1016/j.stem.2020.11.005.

Nakamura, Tomonori, Yukihiro Yabuta, Ikuhiro Okamoto, Kotaro Sasaki, Chizuru Iwatani, Hideaki Tsuchiya, and Mitinori Saitou. 2017. 'Single-Cell Transcriptome of Early Embryos and Cultured Embryonic Stem Cells of Cynomolgus Monkeys'. Scientific Data 4 (1): 487. https://doi.org/10.1038/sdata.2017.67.

Neagu, Alex, Emiel van Genderen, Irene Escudero, Lucas Verwegen, Dorota Kurek, Johannes Lehmann, Jente Stel, et al. 2020. 'in vitro Capture and Characterization of Embryonic Rosette-Stage Pluripotency between Naive and Primed States.' Nature Cell Biology 22 (5): 534-45. https://doi.org/10.1038/s41556-020-0508-x.

Niu, Yuyu, Nianqin Sun, Chang Li, Ying Lei, Zhihao Huang, Jun Wu, Chenyang Si, et al. 2019. 'Dissecting Primate Early Post-Implantation Development Using Long-Term in vitro Embryo Culture'. Science 366 (6467). https://doi.org/10.1126/science.aaw5754.

Xiang, Lifeng, Yu Yin, Yun Zheng, Yanping Ma, Yonggang Li, Zhigang Zhao, Junqiang Guo, et al. 2020. 'A Developmental Landscape of 3D-Cultured Human Pre-Gastrulation Embryos'. Nature 577 (7791): 537-42. https://doi.org/10.1038/s41586-019-1875-y.

*Reviewer #2 (Recommendations for the authors):*

The authors examine the effect of using AE-amniotic epithelium as feeder cells, instead of MEFs, on human primed stem cell. They describe subtle changes in gene expression and claim they influence hESC functionality.

While this topic is important and may have influence on human primed ESC biology and differentiation, there are major caveats in the current version of this paper. Further the paper does not transcend above phenomenology of hESC behavior.

1 – While the authors claim that this makes an alternative new state, they do not convincingly show a totally new marker expression or new functionality. Thus, in my opinion, this does not qualify for a new state. These changes are merely minor expected ones that happen when a certain parameter is changed.

2 – it is often not clear how many lines were used per comparison (gene expression, differentiation outcome etc.). At least 4 different lines should be used to reach meaningful conclusions. Further, isogenic states of lines should be used for comparative analysis when trying to quantify subtle changes and biases.

3 – Will the same changes be observed when using irradiated human fibroblasts as feeder cells.

4 – Can the authors provide any mechanistic insight why these changes occur? what special factors AE cells secrete perhaps?

*Reviewer #3 (Recommendations for the authors):*

In this manuscript Ávila-González et al., analyse the possibility that human amniotic epithelial cells (hAECs) may help to sustain the pluripotency of human embryonic stem cells (ESCs). They find that hESCs cultured with hAECs have normal levels of core pluripotency gene marker expression suggesting that these cells secrete factors that can help promote pluripotency. When analysing the state of pluripotency of hESCs co-cultured with hAESc, they find that they do not express some if the conventional pluripotency markers but neither are they in a naïve state of pluripotency. These cells instead show high expression of phospho-kinases and cytokines related to the mTOR, MAPK and JAK/STAT pathways as well as expression of what the authors define as intermediate pluripotency gene expression. Additionally the cells show low Hox gene expression as well as a more efficient differentiation into anterior neural cell types.

Overall, this is an interesting paper that reports a novel pluripotency culture condition and very possibly the isolation of a new pluripotent cell type that could be useful for the application of stem cells to regenerative medicine. However, there are some points that would benefit from being addressed and these primarily relate to the presentation of the results as well as teasing out the significance of the observations made. These are detailed in the recommendation to authors section.

1. The main point that the authors should clarify the extent to which the gene expression changes they observe in the hESCs co-cultured in with hAECs is coming from the hESCs rather that the hAECs. It is not clear from the text how the hESCs were isolated from the hAECs for their analysis and if they were not how it impacts on the analysis. Similarly, it is not clear the degree to which the expression of genes in hAECs is responsible for the expression of the genes un Figure 6C-D in AMIQ cells co-cultured with hAECs.

2. Much of the RNAseq analysis could be discussed in a more abbreviated manner as given there is little validation performed it is probably best not to overinterpret the results. The manuscript would benefit from a more streamlined description of these results. From this perspective, the authors may not need 2 main figures to present the RNAseq data.

3. Related to the point above, it is not clear what the significance is of many of the RNAseq analysis results shown. There is little attempt at validation or testing their functional significance. Can the cells differentiate more readily into amnion or primordial germ cells, as is speculated to be the case for formative cells?

---

## [Author Response]

Essential revisions:1) Better evidence that these cells represent a new pluripotent state. All three reviewers would like to see a better indication of what defines this state, improved gene expression analysis, in vivo comparsons and analysis of gene expression. What is the profile of hAEC vs hESC in these co-cultures? Two of the reviewers felt that the existing RNAseq analysis takes up large sections of the main figures, but add little to the narrative.

To define the in vitro pluripotent status of hESC-hAECs at the transcriptome level, we compared bulk RNA-seq data vs. other hPSC lines under conventional primed, feeder-free, expanded, and formative conditions. The data was represented in a principal components analysis (new Figure 3A) and in a heatmap (new Figure 3B). On the other hand, we carried out computational deconvolution analysis using CIBERSORT to estimate the relative similarity of the bulk RNA-seq of hESC-hAEC with the scRNA-seq of human epiblast (Xiang et al., 2020) and pre-gastrula embryo lineages (Tyser et al., 2021) (new Figure 3C).

The transcriptomic profile of hAEC vs hESC was compared and the DEG results were visualized in the pathway enrichment analysis by Cytoscape (Figure 3—figure supplement 5B). This new set of analyses were performed by Gerogina Hernandez-Montes and Eliezer Flores-Garza.

In agreement with the reviewers, we moved the RNA-seq analyses of the main figures to supplemental figures. This information includes the number of genes with differential expression (Figure 3—figure supplement 1), the KEGG comparison (Figure 3—figure supplement 2), and pathway enrichment analysis (Figure 3—figure supplement 4).

2) Some evidence that molecule(s) produced by hAEC cells can induce this pluripotent state and/or the mechanism by which they induce it would be essential for publication in eLife.

The proteome arrays suggested an activity of signaling pathways of the SRC kinases, PI3K/AKT, ERK2 and STAT3 in hESC when these are cocultured with hAEC. In a new set of experiments, we separately inhibited each of these signaling pathways with small molecules in the hESC cultures on hAEC and the conventional primed counterpart hESC on iMEF. Inhibitors were added on two consecutive days in hESC medium supplemented with FGF2. Subsequently, we looked for differences in the presence of pluripotency factors and expression of the genes that we detected as upregulated in hESC-hAEC (new Figure 6). In addition, we carried out the neural induction protocol, to investigate whether an initial inhibition of any of these pathways could alter the previous results of increased neural potential in hESC-hAEC. The small molecules used were PI3K inhibitor LY294002 (5 µM and 10 µM), ERK2 inhibitor PD0325901 (1 µM and 2 µM), SRC inhibitor A419259 (1 µM and 3 µM), STAT3 inhibitor Stattic (2.5 µM and 5 µM). We found a differential response to iSRC and iERK2 treatments since the neural differentiation potential decreases in hESC-hAEC in contrast to hESC-iMEF (new Figure 7). Although we used at least two different concentrations for each inhibitor, the qualitative results of the immunofluorescences and the quantitative analyses shown in the figures correspond to the lowest concentration of each inhibitor. This new set of experiments was performed by Carla P Barragan-Álvarez.

3) While the authors have focused on enhanced anterior neural specification, the notion that this is biased or regionalized state of pluripotency, would also suggest that differentiation to other lineages might be restricted. They should compare the relative efficiencies with which hAEC cultured hESCs and standard hESCs differentiate into primitive streak derivatives or primordial germ cells.

In this new version of our manuscript, we carry out the protocol with some modifications of human primordial germinal cell (hPGC) induction reported by Sasaki et al., 2015 and Chen et al., 2019. We compared the efficiency to differentiate hPGC in hESC cultured on hAEC vs standard hESC on iMEF, by detection of germline markers BLIMP1, TFPA2C and STELLA by immunofluorescence. Representative micrographs and quantitative immunofluorescence analyses are in the new Figure 4—figure supplement 1, Figure 4—figure supplement 2 and Figure 4—figure supplement 3.

Reviewer #1 (Recommendations for the authors):González et al., chacterized the hAEC culture condition and found it was dependent on exogeneous FGF2 and then characterized specific signalling pathways there were unique to the hAEC supported pluripotency and cytokines produced by hAECs. In particular they observed an abundance of Src activation in hAEC supported ESCs. They follow this up with transcriptome analysis, where they compare hAEC supported cells to different published ESC culture systems. There are a lot of comparisons, but the bottom line is these cells appear anterior epiblast poised for neural differentiation. While the generation of neural progentintors from all lineages tested was similar, the hAEC cultured cells gave rise to enhanced differentiation towards neural lineages found in the forebrain. They conclude by suggesting hAEC-cultured ESCs exhibit upregulation of polycomb targets and propose that hAEC cells produce a factor that could repress polycomb activity. They suggest that this mechanism might by mediated by miR200b/c miRNA, highly expressed in hAEC and upregrulated in hESCs cultured with hAEC.As suggested above, this paper suggests hAEC impart regionalation or developmental identity on hESC cultures, produces cells that resemble the anterior epiblast that is uniquely poised for anterior neural differentiation. The paper could be strengthened by providing a better assessment of the cell states formed by co-culture with hAECs, how do they cells compare to available primate RNAseq datasets and how homogeneously are they anterior epiblast? In a similar set of analysis they could address the nature of the hAECs used here and explore whether they represent regionalized amnion? In addition the paper suggests candidates for the molecules produced by hAECs that induce this phenotype, but they do not provide any functional evidence for their candidates.As eluded to in the public review, this manuscript introduces a novel idea, that amnionic epithelial cells can instill positional epiblast identity onto human ESC cultures. However, there remain a number of issues that should be resolved prior to publication in eLife.1. The gene expression analysis in Figure 4-6 is incomplete and yet takes up a large part of the main figures. If these cells generally represent regionalized, pre-neural epiblast, normally found adjacent to the amnion, then some analysis of the single cell sequencing from the human and primate embryo data should be undertaken (Xiang et al., 2020; Niu et al., 2019; Nakamura et al., 2017), this analysis should also include the spectrum of ESC cell states that have been published. The authors perform a number of comparisons, but none of these place the hAEC supported cells in context of this spectrum; they could for example compare gene expression changes that occur as development progresses to the gene expression changes that occur in the different ESC cell states as they progress or alternatively try some sort of global analysis of developmental stage with the different ESC data.

In agreement with reviewer 1, the result of the expression analyses of Figures 4 and 5 were moved to the Supplementary Figures of the new Figure 3.

To identify in which part of the spectrum of pluripotency and developmental stage the hESC-hAEC condition can be located, we used scRNA-seq data from cells of the preimplantation and postimplantation epiblast (Xiang et al., 2019) and epiblast and extraembryonic lineages of a pregastrula human embryo (Tyser et al., 2020), as well as the bulk RNA-seq data of different hPSC, both primed, feeder-free, naïve, expanded and formative conditions. The comparison between in vitro hPSC lines is shown in the Figure 3A-B, with a PCA analysis and a heatmap of the matrix signature of the differential expression data. Additionally, we obtained the relative ‘composition score’ of the bulk RNA-seq of hESC-hAEC and of the other hPSC lines vs scRNA-seq of embryo data (Figure 3C).

2. The model in 8b implies that the amnionic epithelium has regionalized inductive properties. What cell type is represented by the hAEC cells. Using similar analysis to that described above, they could address this issue. It also might help them to refine their list of potential “inductive,” molecules.

In the sort of global analysis of RNA-seq to compare hPSC lines with the data embryo, we included the hAEC. Interestingly, hAEC transcriptome have similarity to the extraembryonic mesoderm and non-neural ectoderm (presumptive nascent amnion) (Tyser et al., 2020). To refine the list of potential mechanisms by which hAECs exert their effect on hESC, we focused on the signaling pathways we identified in proteome arrays.

3. There is a lot of discussion of potential candidates for induction of anterior epiblast fate by hAECs, but none of these are tested. To produce the level of advance required for eLife, some demonstration of a candidate produced by the hAEC cells that promotes anterior epiblast identity in ESCs, is important. Even better, if the authors could find a candidate that was also localized in the amnion based on the single cell transcriptome data and that was produced by their hAEC cells this would make the manuscript much stronger.

In this new version of our manuscript, we reported that the inhibition of the SRC and ERK2 pathways produces alterations in the identity of hESC-hAEC condition such as presence and expression of pluripotency factors (new Figure 6 and Figure 6—figure supplement 1), expression of intermediary/epithelializing genes (new Figure 7—figure supplement 1), and in their potential for neural differentiation (new Figure 7).

4. The demonstration that these cells are poised for anterior neural specification is very interesting. Are they less effective at generating primitive streak cell types?

Although we did not evaluate the expression of genes associated with the embryonic mesoderm lineage, specifically from the primitive streak nascent, in the new results we demonstrated that hESC-hAEC have the capacity to express some germline-associated markers, through a protocol in which the induction is via incipient mesoderm-like cells. We found that hAEC-hAEC presented a higher number of TFAP2C+/STELLA+ and BLIMP1+/STELLA+ cells than the standard hESC on iMEF.

5. There appears some confusion in the text about the term pluripotency. It is not really pluripotency that becomes regionalized, but epiblast. Or one could generate pluripotent cells with regional identity. The is also a bit of confusion between different stages of pluripotency. For example, the so called "Rosette," state and "Formative," state. The Rosette state requires only inhibition of Wnt signaling (Neagu et al., 2020), while Formative pluripotency is achieved based on relief of FGF/ERK inhibition as well (Kalkan et al., 2017) and trapped by Wnt and RAR inhibition(Kinoshita et al., 2021), and it is not clear how different "formative," is from some variations of standard, primed human ESCs (Davidson, Mason, and Pera 2015). However, anterior neural differentiation can be directly accessed in the presence of a block to ERK from naïve pluripotency, suggesting it can occur via something like a Rosette state or pre-formative epiblast (Hamilton and Brickman 2014). When and how does anterior epiblast become primed for differentiation away from primitive streak derivatives? This is an excellent question, but the data in the paper doesn't really address this. As it is difficult to resolve so called "formative," markers from anterior epiblast, perhaps it would be easier to avoid overuse of specific labels and stick with analogous embryonic structures. While these cells are FGF2 dependent, it would suggest they are more "primed," rather than formative but as this is all in a human context it is difficult to resolve this and as eluded to above primed pluripotency in some human examples maybe closer to what is called formative. Moreover, I also find it hard to resolve "formative," and anterior epiblast, based on the handle of markers assessed here. Perhaps they will be able to derive better evidence for this staged model of differentiation from the transcriptomic analysis requested in point 2, but I am not sure this will provide a clear answer. Experimentally they could ask how dependent these cells on ERK signaling? This would be interesting to resolve, as ERK activity is significantly lower in the anterior epiblast than on the posterior side in the region of the primitive streak. They could attempt to convert naïve cells to this state through a staged differentiation trajectory with specific inhibitors and demonstrate when these cells become uniquely anteriorized? While these are all interesting experiments, they could also just revise the discussion and Figure 8 and focus on epiblast-like, anterior epiblast-like.

We agree with the reviewer that pluripotency is not regionalized, but rather cells that have the pluripotent state (for example, formative or primed) are the ones that segregate (for example, in the anterior or posterior part of the epiblast). We also agree with the reviewer that there is currently no consensus to define the formative stage in either the human embryo or in vitro culture. Regarding the possible role of ERK in this condition, we detected by immunofluorescence an increase of ERK2 in hESC-hAEC compared to hESC-iMEF. Therefore, we inhibited ERK2 with PD0325901, and found that it alters the identity of hESC-hAEC, as mentioned in the answer to question 3. It would be interesting and enlightening to study the transition of naïve hPSC when cultured in interaction with the amniotic epithelium to identify whether the anterior epiblast-biased formative/primed state is obtained; these experiments would be part of a future publication.

Specific Comments1. Otx2 upregulated in AEC feeder culture, but not with conditioned media (Figure 2B)? As Otx2 is one of the principle markers interpreted, perhaps the authors should give this some thought.

From the observation of OTX2 upregulation (old Figure 2B is now Figure 1G), we began to interpret that hESC-hAEC could potentially have a neuroectoderm bias. We assume that the biased effect is attributed to the direct interaction with the amniotic epithelium, since the upregulation of OTX2 was not observed in the conditioned media.

2. Increase size of labels on the figures, they are very difficult to read.

We increased the size of figures and letters as suggested by the reviewer.

3. hAECs have some pluripotent gene expression and qualities. What is the extent that the data in Figure 3C, reflects the activity in hAECs or hESCs? Should these experiments not be using conditioned media?

We did previous immunofluorescences to detect STAT3, ERK2 and AKT in the hESC-hAEC-CM (conditioned medium) condition and no marks were observed (data shown) as in coculture hESC-hAEC.

Reviewer #2 (Recommendations for the authors):The authors examine the effect of using AE-amniotic epithelium as feeder cells, instead of MEFs, on human primed stem cell. They describe subtle changes in gene expression and claim they influence hESC functionality.While this topic is important and may have influence on human primed ESC biology and differentiation, there are major caveats in the current version of this paper. Further the paper does not transcend above phenomenology of hESC behavior.1 – While the authors claim that this makes an alternative new state, they do not convincingly show a totally new marker expression or new functionality. Thus, in my opinion, this does not qualify for a new state. These changes are merely minor expected ones that happen when a certain parameter is changed.

In agreement with reviewer 2, it is difficult to identify a specific marker for a delimited pluripotent state since human pluripotency during development is rather a continuum. Instead of considering it as a new additional pluripotent state within the naive-formative-primed spectrum, we could consider it a substate or subpopulations between the formative and primed pluripotency that would segregate to the anterior neuroectoderm. Embryonic data analyses have not yet identified anterior-posterior areas of the epiblast, so we rely on the expression of anterior neuroectoderm markers such as LHX5, SIX3, and the superior potential to differentiate to neural fate under hAEC conditions.

2 – It is often not clear how many lines were used per comparison (gene expression, differentiation outcome etc.). At least 4 different lines should be used to reach meaningful conclusions. Further, isogenic states of lines should be used for comparative analysis when trying to quantify subtle changes and biases.

Pluripotency characterization experiments (marker detection by immunofluorescence, expression of specific genes by RT-qPCR, etc.) were carried out under different conditions (on iMEF and hAEC feeder layer, and with iMEF conditioned medium and hAEC conditioned medium), using three lines: H9, H1 and Amicqui- 1. We previously studied a fourth hESC line (‘in house’ Amicqui-2, Avila-Gonzalez et al., 2018) in which no differences were detected between conditions (data not shown in this article). For functional analyses of differentiation capacity (germline specification and neural induction), we used at least the Amicqui-1 and H9 lines under iMEF and hAEC conditions, but quantitative analyses were done only on the Amicqui-1 line for isogenic comparison.

3 – Will the same changes be observed when using irradiated human fibroblasts as feeder cells.

Human fibroblasts will have the same effect as iMEF as a feeder layer because both have a mesenchymal phenotype, unlike hAECs which have an epithelial phenotype, which would decrease the epithelial-mesenchymal transition in hESCs when grown on a fibroblastic feeder layer.

4 – Can the authors provide any mechanistic insight why these changes occur? what special factors AE cells secrete perhaps?

As suggested by the reviewer, we used small molecules to selectively inhibit SRC, PI3K/AKT, ERK2, and STAT3 signaling pathways in hESC-hAEC coculture. In the new modifications, we suggest that SRC family kinases activate other pathways such as ERK2 and STAT3, through which it induces the biased neural potential of hESC. The SRC pathway can be activated either by focal adhesions (Figure 3—figure supplement 2) or via the EGFR pathway (Figure 2A) in hESC-hAEC.

Reviewer #3 (Recommendations for the authors):In this manuscript Ávila-González et al., analyse the possibility that human amniotic epithelial cells (hAECs) may help to sustain the pluripotency of human embryonic stem cells (ESCs). They find that hESCs cultured with hAECs have normal levels of core pluripotency gene marker expression suggesting that these cells secrete factors that can help promote pluripotency. When analysing the state of pluripotency of hESCs co-cultured with hAESc, they find that they do not express some if the conventional pluripotency markers but neither are they in a naïve state of pluripotency. These cells instead show high expression of phospho-kinases and cytokines related to the mTOR, MAPK and JAK/STAT pathways as well as expression of what the authors define as intermediate pluripotency gene expression. Additionally the cells show low Hox gene expression as well as a more efficient differentiation into anterior neural cell types.Overall, this is an interesting paper that reports a novel pluripotency culture condition and very possibly the isolation of a new pluripotent cell type that could be useful for the application of stem cells to regenerative medicine. However, there are some points that would benefit from being addressed and these primarily relate to the presentation of the results as well as teasing out the significance of the observations made. These are detailed in the recommendation to authors section.1. The main point that the authors should clarify the extent to which the gene expression changes they observe in the hESCs co-cultured in with hAECs is coming from the hESCs rather that the hAECs. It is not clear from the text how the hESCs were isolated from the hAECs for their analysis and if they were not how it impacts on the analysis. Similarly, it is not clear the degree to which the expression of genes in hAECs is responsible for the expression of the genes un Figure 6C-D in AMIQ cells co-cultured with hAECs.

For gene expression evaluation experiments either by RNA-seq or RT-qPCR, hESC on hAEC were detached with Accutase. We have observed that an enzymatic treatment with Accutase for 5 min at 37°C only detaches the hESC, while the hAEC remain adhered to the plate. We did STR analysis at various passages to confirm that Amicqui-1 or another hEASC line was not contaminated with any other cell type of human origin.

2. Much of the RNAseq analysis could be discussed in a more abbreviated manner as given there is little validation performed it is probably best not to overinterpret the results. The manuscript would benefit from a more streamlined description of these results. From this perspective, the authors may not need 2 main figures to present the RNAseq data.

We appreciate the reviewer´s suggestion. We moved the main Figures 4 and 5 from RNA-seq analysis to supplemental images (Figure 3—figure supplement 1-5)

3. Related to the point above, it is not clear what the significance is of many of the RNAseq analysis results shown. There is little attempt at validation or testing their functional significance. Can the cells differentiate more readily into amnion or primordial germ cells, as is speculated to be the case for formative cells?

In the new set of experiments, hESCs maintained on either hAEC or iMEF were challenged to a germline induction protocol, as reported by (Sasaki et al., 2015, Chen et al., 2019). The results are shown in Figure 4—figure supplement 1, Figure 4—figure supplement 2 and Figure 4—figure supplement 3.